# New internationalization paths of Chinese brands: A configurational study

**Junfeng Liao**[1,2]*, **Minru Yang**[1]

**1** Department of Electronic Business, South China University of Technology, Guangzhou, China, **2** School of Economics and Management, Kashi University, Kashi, China

* ljf@scut.edu.cn

## Abstract

Although several new international brands in the information technology services industry have emerged in China, most previous research on the paths of internationalization taken by Chinese brands has focused on the manufacturing industry. Further exploration of the diversity and novel characteristics of these paths remains necessary. Taking into account the different industries and the interactions among multiple factors that are relevant in this context, supplementary research on the paths of internationalization taken by Chinese brands is also needed. Simultaneously, more research on the mechanisms underlying brand empowerment during the internationalization process is necessary. Based on a theoretical framework for brand empowerment, this article analyzes 61 representative Chinese brands using fuzzy set qualitative comparative analysis (fsQCA). This article reveals the following main findings: (1) brand penetration and brand acculturation are two key capabilities with regard to brand internationalization; (2) two modes of brand internationalization are evident in China, i.e., progressive internationalization and leapfrog internationalization; (3) four paths of internationalization can be observed with regard to Chinese brands, including two new paths, i.e., L-S-Cu and P-S-M; and (4) six additional paths are worthy of further exploration.

**Data Availability Statement:** All relevant data are within the paper and its Supporting Information files.

**Funding:** This work was supported by the National Social Science Fund of China (18BGL110). The funders provided financial support and had no role

## 1. Introduction

Due to the implementation of China's "Going Global" strategy, the pace of internationalization among Chinese brands has accelerated. In 2022, 45 Chinese brands were included in "The World's 500 Most Influential Brands", the fourth highest ranking of any nation. While this number has doubled from 2013 (25), it is still only one-fifth as high as that the comparable figure in the US (198). This fact indicates that China is still a long way from becoming a large country in terms of brands.

According to Kantar's 2022 Chinese Global Brand Builders, which is shown in Table 1, many excellent information technology service industry overseas brands have emerged in China. Among the selected brands, those that are less than 10 years old account for 32% of the total, including representative brands such as ByteDance (Platform/App) and miHoYo (Game). We observe that the focus of such brands in terms of industry has shifted from

in study design, data collection and analysis, decision to publish, preparation of the manuscript.

**Competing interests:** The authors have declared that no competing interests exist.

consumer electronics (28%) and home appliances (10%) to the information technology service industry (38%). Although research on the internationalization of Chinese brands has increased, previous research has mainly focused on manufacturing industries such as consumer electronics [1] and home appliances [2]. Due to the emergence of many international brands in the information technology service industry, the internationalization paths taken by Chinese brands have exhibited new characteristics due to the brands' heterogeneity in terms of industry [3]. The diversity of the internationalization paths taken by Chinese brands requires further exploration [4].

This study investigates 61 representative Chinese brands, including brands in the manufacturing and service industries, with the goal of exploring the following question: "In the new context, what new paths for the internationalization of Chinese brands will emerge?". The innovative contribution of this article lies in the fact that we identify four paths for the internationalization of Chinese brands, two of which support the claims of previous research [1, 5]. However, we also reveal two new single-capability paths, which represent important supplements to previous research [4, 6]. Furthermore, through in-depth analysis of the paths, we provide some suggestions for choosing paths for brand internationalization across different industries, which differ from the recommendations of previous studies. Additionally, six other logical paths that are not covered by the research sample on which this article focuses are discovered, providing a reference and suggestion directions for future research.

## 2. Literature review

The internationalization of Chinese brands is mainly divided into three stages: "go global", "be local" and "lead global". "Go global" refers to the task of allowing the world to become familiar with the brand. This stage focuses mainly on the transnational management problems that emerge during the early stages of the brand's global expansion [2, 7–11]. "Be local" indicates that the brand is becoming a mainstream brand in the host country. At this stage, the main concern is the legitimacy of the brand in the host country [2, 12–16]. "Lead global" indicates that the brand is becoming a truly global brand. At this stage, the main focus is on comprehensive capability improvement and brand internationalization performance [17–21]. The literature review is divided into three stages and presents key information in the form of a table (see Table 2).

**Table 1. Chinese global brand builders 2022.**

| Industries | Brands | Number | Proportion | Number of young brands | Proportion of young brands |
|---|---|---|---|---|---|
| Platform/App | ByteDance, DIDI, KunLun, KuaiShou | 4 | 38% | 2 | 32% |
| E-commerce | Alibaba, Shein, JD, Light in the box | 4 | | 0 | |
| Game | Tencent, Funplus, miHoYo, Lilith, 37Games, IGG, Century Games, NetEase, Zenjoy, IM30, Magic Tavern | 11 | | 4 | |
| Consumer electronics | Lenovo, Xiaomi, Huawei, OPPO, Oneplus, Vivo, Anker, Dji, ZTE, honor, TP-Link, Aukey, Ecoflow, Realme | 14 | 28% | 4 | 29% |
| Home appliances | Hisense, Haier, TCL, Ecovacs, Midea | 5 | 10% | 0 | 0 |
| Automotive | BYD, GWM, Chery, Geely, JAC | 5 | 10% | 0 | 0 |
| Other | Tsingtao Beer, air China, China Eastern, Bank of China, ICBC, WORX, FlexiSpot | 7 | 14% | 0 | 0 |
| Sum | | 50 | 100% | 23 | - |

Notes: This research defined young brands as those that were less than 10 years old in 2022. It evaluated the proportion of young brands among selected brands within each industry.

Data source: Kantar BrandZ Top 50: Chinese Global Brand Builders 2022 report.

**Table 2. Literature on the three stages of Chinese brand internationalization.**

| Internationalization stages | | Representative literature | Research problems | Research objects | Research methods | Theoretical support |
|---|---|---|---|---|---|---|
| Single stage | Go global [2, 7–11] | [22] | This study investigates the impact of the ownership of Chinese enterprises on their internationalization, finding that due to the heterogeneity of ownership, enterprises tend to use different internal resources to promote internationalization | Mixed state-owned enterprises | Quantitative (multiple regression) | Institutional theory and resource dependence theory |
| | | [23] | This study explores the competitive scenario of EMNE, including the relationship between the allocation of awareness-motivation-capability (AMC) conditions and the comparative institutional advantage of its strategic asset-seeking destination | | Qualitative & quantitative (fsQCA) | The resource-based view and institutional theory |
| | Be local [2, 12–16] | [15] | This study explores the relationships among institutional distance, brand legitimacy, and brand internationalization performance and identifies the acquisition strategies and boundary conditions associated with brand rationality | High-tech enterprises | Qualitative (theoretical summary) | Institutional theory |
| | Lead globally [17–21] | [24] | The key antecedents of relationship performance in small and medium-sized enterprises (SMEs) are examined from the perspectives of organizational capability and organizational learning, indicating that relationship embeddedness and relationship memory have positive impacts on relationship performance | 223 internationalized Chinese SMEs | Quantitative (multiple regression) | Social network theory and organizational learning theory |
| | | [25] | The article clarifies the paradoxical relationship between "brand isomorphism" and "brand heterogeneity" as well as the nodes and boundary conditions associated with the impacts of these factors on brand overseas performance | Consumer research | Quantitative (experimental method) | Institutional theory |
| Multistage | | [26] | This study investigates the relationship between internationalization paths and corporate ownership using correspondence analysis, thereby exploring the paths taken by Chinese enterprises as part of the internationalization process and the preferred destinations for foreign direct investment | Chinese manufacturing enterprises | Quantitative (correlation analysis) | |
| | | [4] | A conceptual framework is proposed to compare the roles of springboards across three commonly used upgrade paths: path-following, path-compressing, and path-creating. | - | Qualitative (theoretical) | Springboard theory |
| | | [6] | This study uses TTF jewelry brand as a case study object to explore the brand internationalization path and the underlying mechanism from the perspective of empowerment. | TTF premium jewelry brand | Qualitative (single case study) | Brand empowerment and organizational learning theory |
| | | [1] | From the perspective of patent ecological operation capability, this study analyzes the relationships among patent ecological operation capability, brand empowerment, and brand internationalization and explores the brand internationalization path and corresponding growth mechanism in the context of intelligent manufacturing enterprises | Huawei and Xiaomi | Qualitative (dual case study) | Brand empowerment and boundary crossing |
| | | [5] | This study constructs a framework of "internationalization context-strategic selection-enterprise performance" to reveal the institutional-resource dual context faced by emerging economy enterprises in the internationalization process and proposes an internationalization strategy that matches the dual context to ensure sustained legitimacy and performance. | ZTE, Jinli, and Tianlong | Qualitative (multicase analysis) | Institutional theory and the resource-based view |

In single-stage brand internationalization research, the "go global" stage focuses on transnational management problems during the early stages of the brand's global expansion, such as those pertaining to driving factors [27], site selection [22] and entry strategies [23]. Some such studies have used empirical methods based on institutional theory and resource dependence theory to explore how the ownership structures of Chinese enterprises influence going global [22] and found that due to heterogeneous ownership, enterprises utilize different internal resources to promote internationalization. Additionally, the complex interactions between internal resources and external institutional environments drive enterprises to expand globally. Other studies have examined site selection from a resource-seeking perspective [23]. Leveraging resource-based views and institutional theory, the findings of such studies have highlighted the joint influence of internal awareness factors and external motivational factors on site selection decisions. Such research has revealed that Chinese high-tech enterprise brands primarily seek market and technology resources when expanding globally. In terms of "be local", researchers have mainly examined the legitimacy issues faced by brands in the host country [25]. Strategies to promote legitimacy in host countries have been analyzed based on institutional theory. The institutional environment shapes consumer legitimacy perceptions and simultaneously influences the operating rules followed by brands embedded in host markets [15]. Differences in hosts in terms of institutions, markets and cultures can generate pressures, indicating that alignment with local environments is necessary for mainstream success. "Lead global" indicates that the brand is becoming a truly global brand. This stage focuses on brand capability enhancement and global performance. A study that applied social network and organizational learning theory in this context found positive relationships among embeddedness, absorptive capacity and brand relational performance, thus emphasizing the importance of interorganizational relationships with regard to internationalization [24]. Other research that has relied on institutional and differentiation competition theories to conduct consumer surveys and experiments has discovered that brand homogeneity/heterogeneity impacts foreign brand performance through legitimacy perception mechanisms [25].

Multistage brand internationalization research has focused primarily on the internationalization paths of brands. Some scholars have defined four types of paths: exports to target countries, licensing agreements with foreign firms, transferring technology or through technology cooperating with foreign firms [26]. Other researchers have categorized paths for emerging market multinationals as path-following, path-compressing, or path-creating [4]. Keys factors associated with the upgrading path include host institutional environments in terms of politics, society and economics as well as firm characteristics. The finding that springboard theory is not suitable for all paths emphasizes the fact that the paths associated with homogeneous emerging markets according to the traditional view are also diverse. Further exploration is needed to explore the diversity of internationalization paths taken by enterprises in China [4]. One study that used a luxury jewelry brand as a case study proposed the concept of brand empowerment and examined brand internationalization from an empowerment perspective, discovering that for emerging market companies, brand internationalization can follow a "leapfrog" type of path. Additionally, this study highlighted the fact that a brand's internationalization capabilities may evolve dynamically over the course of a company's global life cycle [6]. Another study analyzed how patent ecosystem operation capabilities, brand empowerment and internationalization are interrelated in the context of smart manufacturers, revealing two paths: technology-market-management and market-technology-management [1]. An examination of successful Chinese smartphone internationalizers based on a contextual-strategic-performance framework revealed institutional-resource dual contexts. This research revealed that Chinese companies face a dual institutional-resource context during internationalization [5]. It emphasized the fact that the choices associated with Chinese brand internationalization

strategies are mutually determined based on the interaction between enterprise resource advantages and institutional distance factors. Furthermore, this research proposed matching international market entry strategies and host country brand strategies to the contextual situation, thereby helping achieve performance outcomes.

An examination of single-stage internationalization research thus reveals that previous work has often adopted a corporate perspective to study the challenges that companies face during specific stages of internationalization [15, 28]. In addition, issues pertaining to brand internationalization have received less inquiry. However, the fact that single-stage studies have tended to rely on institutional theory and the resource-based view, emphasizing the fact that decisions [23], strategies [4] or performance [12] result jointly from internal and external influences, is worth noting. The interaction effects among different factors as well as the self-impacts of individual factors on brand internationalization decisions and performance, are equally important issues that deserve examination. Previous research has failed to examine the impacts of the interactions among different factors on brand internationalization [15, 22–24]. As Table 2 shows, research on multistage internationalization paths has been relatively rare, and very few studies have investigated internationalization paths from the perspective of brands [6]. Early exploration of internationalization paths continued to focus on the entry path taken by enterprises [26] rather than the multistage internationalization path taken by brands. Recently, several studies on internationalization paths have focused on different stages of internationalization, emphasizing the fact that the internationalization of enterprises is influenced by the interaction between resources and the environment [3]. The internationalization process of enterprises is also a process that involves continuously empowering brands with internationalization capabilities [6]. The focus of empowering brand internationalization can vary depending on the different life cycles of enterprises [1]. The brand empowerment mechanisms underlying enterprise internationalization remain underexplored. The empowerment mechanism underlying brand internationalization requires further clarification [6]. Simultaneously, previous research has focused mainly on the manufacturing industry, but the internationalization path taken by brands is influenced by the heterogeneity of the industry and thus exhibits diversity [3, 4]. The diversity of the internationalization paths taken by Chinese brands also needs further exploration [4]. In summary, further research on brand internationalization paths that takes into account different industries and the interaction effects among multiple factors is necessary, and the brand empowerment mechanism underlying brand internationalization paths also requires further exploration.

## 3. Research and data methodology

### 3.1. Research model

**3.1.1. Research framework.** This paper defines brand internationalization from three perspectives, i.e., brand equity [29, 30], geographical location, and dynamic process [31]. The paper also maintains that brand internationalization refers to the process of building brand equity in foreign markets, especially in mainstream foreign markets. Brand empowerment is the process by which enterprises increase brand equity through empowerment [1, 32].

Brand internationalization is a dynamic process that is impacted by a variety of factors, including a firm's own resources and capabilities, differences between the firm's domestic/host markets, and industry conditions [4]. The resource-based view posits that brands are shaped by firms' resources and capabilities. The resources and capabilities that companies utilize include intangible assets [1], human capital [5] and relational networks [18, 20]. When companies enter foreign countries, they first encounter host country institutional environments that include governance structures, social norms and cultural value systems [15]. On the one hand,

the institutional environment (technological level, market norms) affects the operational rules applied to the embedded organizations [10, 33]; on the other hand, the institutional environment (cultural system) affects consumers' definition of legitimacy [15]. Brands that enter new international markets often face significant changes in technological level, market norms, and cultural awareness, leading to significant institutional pressure on their internationalization operations. Therefore, an enterprise's resource/capability system must match the institutional environment of the host country [10]. Based on the resource-based view (RBV) and institutional theory as well as research on brand internationalization, this article argues that brand internationalization is driven by both internal factors (the capabilities of the enterprise) and external factors (the institutional environment of the host country).

Brand internationalization is a dynamic process, as part of which internal and external factors drive brand internationalization; it is also the process of brand empowerment [1, 6]. The World Bank defines empowerment as the process of increasing individual or group assets. Scholars have investigated different constructs pertaining to empowerment, such as data empowerment [34] and network empowerment [18], which aim to enable the target of the empowerment to obtain greater capabilities using certain means or methods, thereby accelerating the achievement of goals and creating greater value [1, 6]. Therefore, this study suggests that the process of enhancing brand equity is also a process that involves empowering the brand [1, 32] and that the process of brand internationalization is also a process that entails brand empowerment.

This paper proposes a brand internationalization empowerment framework based on the A-B-C paradigm [35]: antecedents (external/internal drivers)—behavior (empowerment process)—consequence (empowerment result). This model is shown in Fig 1.

**3.1.2. Brand empowerment mechanism.** According to boundary spanning theory, organizations rely on boundary spanners to facilitate smooth social and economic communication between organizations and their external environments, thereby ensuring that organizations are not affected by destructive factors in the external environmental. Brands face multiple boundaries in the process of internationalization, mainly including external and geographical boundaries [36]. This paper posits that the process of brand internationalization is mainly associated with the technological boundary, the market boundary and the cultural boundary [1, 6]. A boundary marks the place at which an organization ceases and its environment starts [37, 38], which is associated with a "screening effect" [1]. As different brands enter different markets, they face different boundary conditions (i.e., the technology, market, and cultural boundaries) [39], and the enterprise's capabilities (technological innovation capability, market expansion capability, and relational embeddedness [40]) have a regulatory effect on the boundary [41]. Therefore, the capability to span each boundary is the boundary spanner [42], which serves as the internal driver, in which context the relevant environment faced by the brand

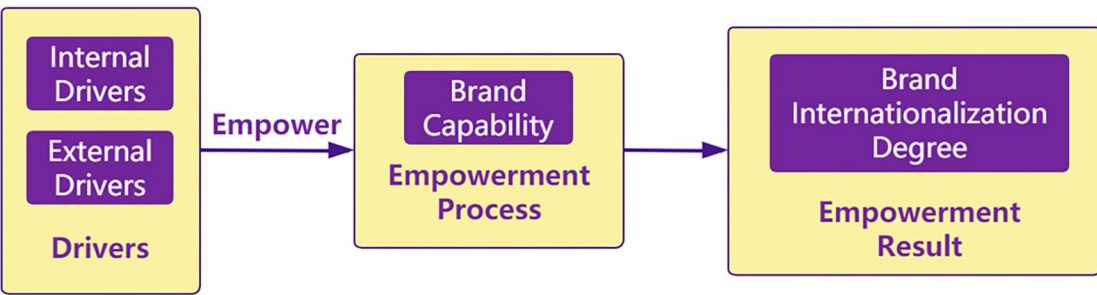

**Fig 1. Research framework for brand internationalization.**

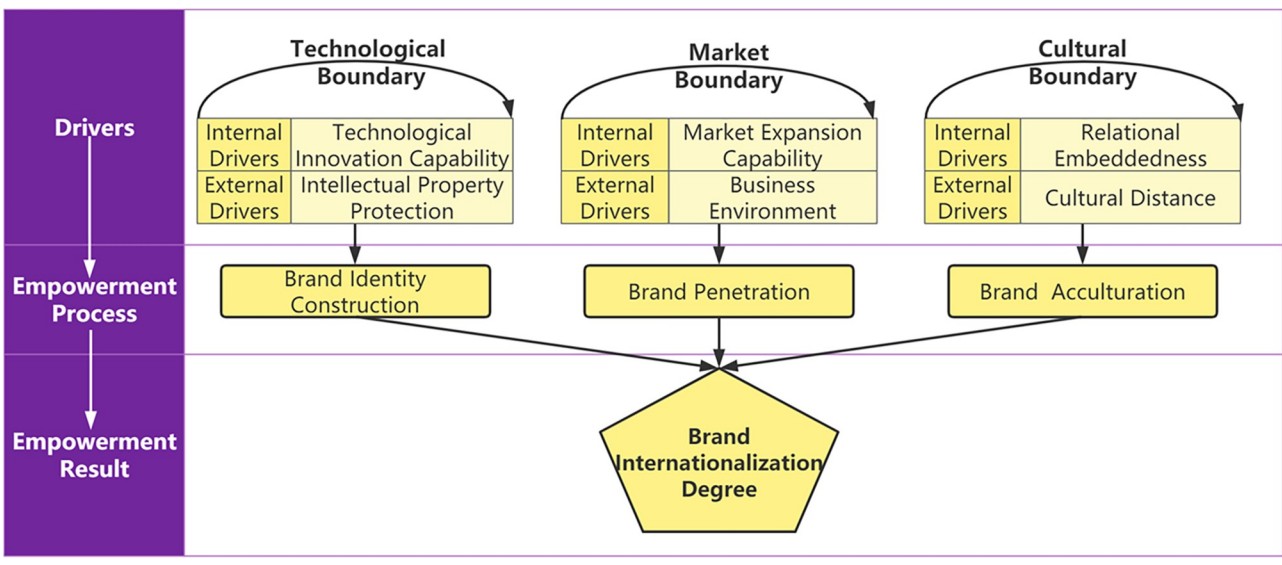

**Fig 2. Research model for brand internationalization.**

represents different boundary situations. Spanning different boundaries can empower brands with different capabilities. Spanning the technological boundary, the market boundary and the cultural boundary can empower brands with the capabilities of identity construction, penetration and acculturation [43]. The interactions among these three capabilities empower the brand to achieve internationalization. The theoretical framework of brand internationalization empowerment is shown in Fig 2.

First, brand identity construction is the core motivation underlying the process of brand internationalization [6, 27]. The key to constructing a brand's identity is to distinguish the brand in question from its competitors and give consumer commitment. Innovation is an important promise that a brand offers to consumers and an important capability that enables the brand to distinguish itself from its competitors [44]. Therefore, this paper claims that brand innovation capability can enable the brand to overcome the "screening effect" of the technological boundary. The interaction between technological innovation capability (TI) and intellectual property protection (IP) empowers the brand with identity construction capability, thus enabling the brand to span the technological boundary [45].

Second, brand penetration represents the brand's important capability to cope with the liabilities resulting from its origin and foreignness [46] The brand's market expansion capability can reflect its penetration capability [11]. The interaction between market expansion capability (ME) and the business environment (BE) of the host country empowers the brand with penetration capability, enabling it to span the market boundary.

Third, brand acculturation is an important capability that enables the brand to overcome cultural differences, gain consumers' trust and exhibit stable development in the host country [6]. According to social network theory, relationship embedding emphasizes the binary trade relationship directly established by the company and refers to the degree of mutual understanding, trust, and commitment [20]. Relational embeddedness represents an important connection between brands and consumers and serves to embody the trust between brands and customers [24, 47]. The interaction between relational embeddedness (RE) and cultural distance (CD) with regard to the relationship between the host country and China empowers the brand with acculturation capability, thus enabling it to span the cultural boundary.

## 3.2. Research method

The current research on Chinese brand internationalization reveals that both qualitative and quantitative methods have been used in single-stage studies, but multistage brand internationalization path studies have focused mainly on qualitative methods. Traditional qualitative research methods (such as case analysis) can focus on one or several research objects and thus develop a relatively good theoretical framework and provide a reference for subsequent research. However, the generalizability of the conclusions must be enhanced using quantitative methods [48]. Although traditional empirical studies such as multiple regression can overcome the problem of generalizability, their analysis focuses on the net effect of a single influencing factor on the outcome variable; however, brand internationalization is determined by multiple factors, such as the capabilities and the host countries' environment as well as the interactions among these factors [9]. Traditional empirical studies cannot effectively address the relationship between multiple causal and interactive effects.

In a single-stage internationalization study, one study uses fuzzy set qualitative comparative analysis (fsQCA) to study the impacts of the interactions among awareness, motivation, and capability on the selection of internationalization resource-seeking destinations by enterprises [23]. This approach provides a direction for this article's attempt to study the causal complexity and multiple causal concurrency characteristics of brand internationalization paths. In light of the advantages of the QCA method with regard to studying multiple concurrent causal relationships by reference to small samples and cases and combining quantitative and qualitative methods of analysis to bridge the gap between quantitative and qualitative research in most fields [34], this article employs the qualitative comparative analysis method fsQCA to study the brand internationalization paths taken by representative Chinese brands. The QCA research method is based on set theory and Boolean logic, and it aims to compare and analyze research cases fully; furthermore, it is suitable for studying multiple concurrent causal relationships across cases. "Multiple" refers to the use of more than one path to facilitate brand internationalization; that is, multiple paths can reach the same endpoint. "Concurrent" indicates that different paths are formed as a set by the interaction of different factors, in which context path sets are related but also distinct from each other [49]. However, the process of internationalization of Chinese brands is relatively short, and relatively few relevant data have been collected. The QCA research method has relatively low requirements with regard to the number of samples, thus making it suitable for studies feature a small sample size. QCA mainly includes three analytical techniques, namely, csQCA (binary variable), mvQCA (multivalued variable), and fsQCA (continuous variable). As the research data referenced in this article are continuous data, fsQCA is used for the study [50].

## 3.3. Research data

This research is based on two authoritative lists from 2018 and 2019, i.e., "The World's 500 Most Influential Brands", which was released by the World Brand Lab, and "Top 50 Leading Chinese Brands", which was released by KPMG and Facebook; a total of 61 brand samples whose main business model is B2C were selected. The data sources of the research variables are shown in Table 3.

Technological innovation capability is defined in terms of the number of patent applications made by enterprises. As multinational enterprises must apply for patents in multiple countries, the same/similar patent applications are submitted in multiple countries. Therefore, when retrieving the number of patent applications, a simple patent family merge is employed.

As interaction with customers is an important manifestation of the close relationship between enterprises and customers [51], brand relational embeddedness is defined in terms of

**Table 3. Research data.**

| Variable | Indicator Quantification | Data Source |
|---|---|---|
| TI | Number of patent applications made by the enterprises | IncoPat global patent database |
| IP | Average of intellectual property protection in major markets that the brand enters | World Economic Forum: "The Global Competitiveness Report 2019" |
| ME | Number of countries that the brand enters | Official corporate website and annual report |
| BE | Average of the ease of doing business in the major markets that the brand enters | World Bank: "Doing Business 2020: Comparing Business Regulation in 190 Economies" |
| RE | Number of followers of the enterprise's Twitter account | Twitter |
| CD | Average cultural distance between China and the major countries that the brand enters | Geert Hofstede official website |
| BI | Average of the normalized value of enterprise's total overseas revenue and the proportion of the enterprise's overseas revenue | Official corporate website and annual report |

Note: BI: Brand internationalization degree

the number of followers of the enterprise's social media account on Twitter. If the enterprise has multiple official accounts, the account with the largest number of followers is selected.

Intellectual property protection, business environment and cultural distance all help determine the market that the brand enters first and help the brand select the major markets that it enters; this measure generally includes the top five countries that the brand enters. The main markets of the sample brands are shown in Fig 3. The sizes of the nodes, labels and edges are all adjusted based on the degree of the nodes, and the nodes with a degree greater than 10 are colored yellow. Fig 3 shows that the main markets of Chinese brand are North and South America, Europe, East Asia and South Asia, while the strength of those brands in the African market and the Middle East market is relatively weak.

Cultural distance is measured in terms of six dimensions [52]. The calculation formula references [53], as shown in Eq (1).

$$CD_j = \sum_i [(I_{ij} - I_{iCH})^2 / V_i] / 6 \tag{1}$$

$CD_j$ represents the cultural distance between host country j and China, $I_{ij}$ represents the value of host country j in dimension i, $I_{iCH}$ represents the value of China in dimension i, and $V_i$ represents the variance of all host country values in dimension i.

The brand internationalization degree is based on the average of the normalized value of the overseas revenue of the enterprise, in which context t is the proportion of the enterprise's overseas revenue [54].

Since the fsQCA method is based on set theory, each variable is considered to be a set; thus, we must calibrate the original data to determine the membership score of each case in the corresponding set. The direct calibration method is more commonly used for such calibration, i.e., calibrating by reference to the sample data's quantile or the natural cutoff point of the sample data. Theoretical calibration, i.e., calibration by extant theory, is also employed [55]. The calibration requirements are as follows: (1) do not truncate unrelated variations, and (2) the calibrated data should be hierarchical [56]. Based on the requirements listed above, we calibrate by reference to the quantile of the sample, i.e., 0.95, 0.5, 0.05, and then make adjustments according to the natural cutoff point and extant theory, as shown in Table 4.

According to springboard theory, for the purpose of institutional arbitrage, brands choose host countries that feature better business environments than the home country [57].

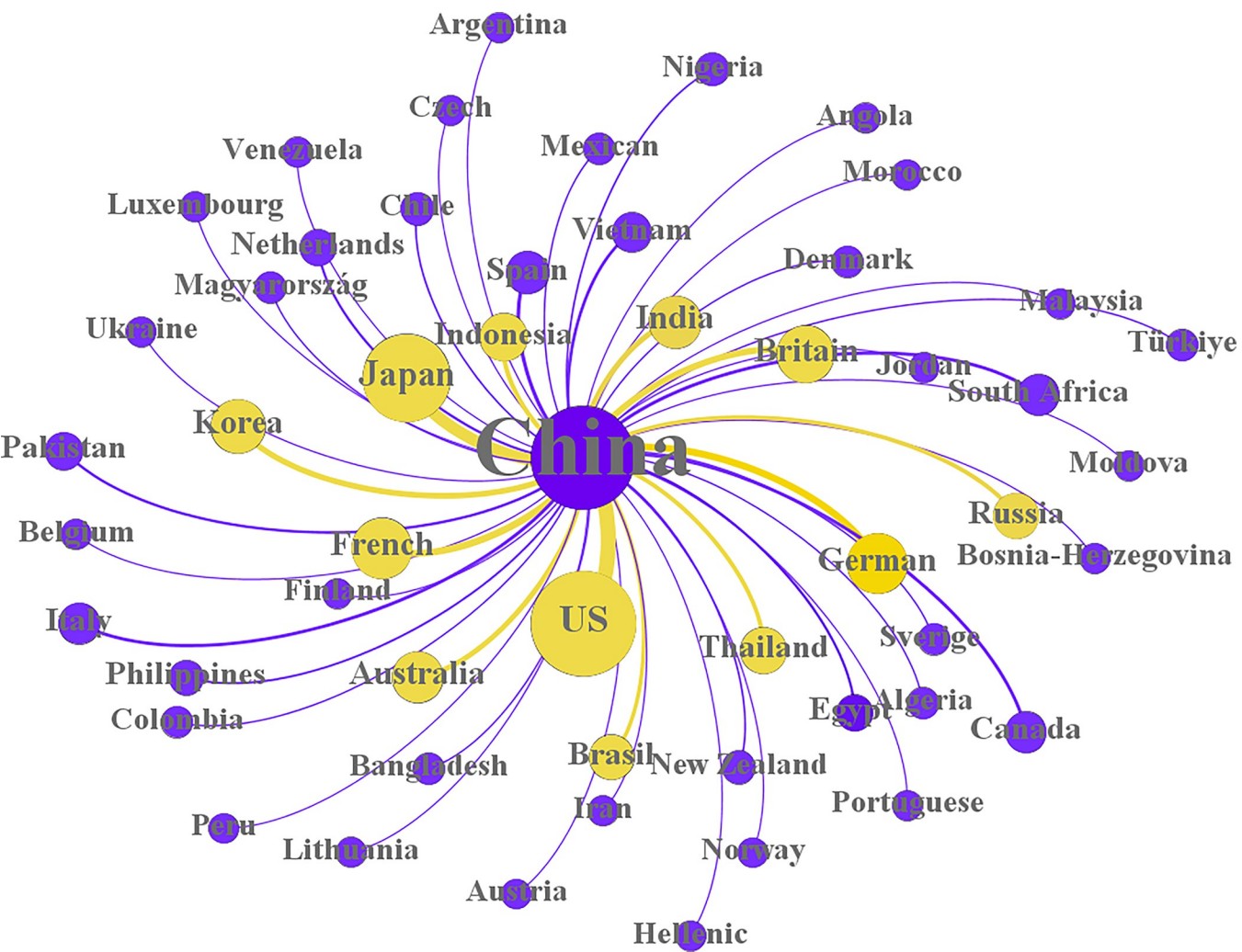

**Fig 3. Network diagram of the major markets of brands.**

Therefore, the calibration results should distinguish between positive environments and negative environments. The maximum ambiguity point of BE should be 77.9 (the data regarding China). Wei claimed that a brand with more than 10% overseas business is an international brand [54]. Therefore, the maximum ambiguity point of BI is 10.

**Table 4. Data calibration.**

| Variable | Calibration | | |
|---|---|---|---|
| | "Full-In" | Maximum Ambiguity | Full-Out |
| TI | 35000 | 600 | 2 |
| IP | 5.7 | 5 | 4.3 |
| ME | 200 | 124 | 10 |
| BE | 81.4 | 77.9 | 68 |
| RE | 730000 | 40000 | 45 |
| CD | 3.6 | 2.7 | 1.7 |
| BI | 50 | 10 | 1.3 |

## 4. Results

### 4.1. Analysis of necessary conditions

A necessity condition is a superset of the result set, while a condition configuration is a subset of the result set [49]. The existence of necessary conditions affects the sufficiency analysis [49]. Therefore, it is necessary to perform the necessity analysis before performing the sufficiency analysis. The necessity of conditions is mainly assessed in terms of consistency. When the consistency is higher than 0.9, the condition is a necessary condition [55]; the formula for consistency is shown in Eq (2).

$$Consistency(Y_i \leq X_i) = \sum(min(X_i, \ Y_i))/\sum(Y_i) \tag{2}$$

As shown in Table 5, the consistency of the conditions is lower than 0.9; thus, there are no necessary conditions for brand internationalization.

### 4.2. Analysis of sufficient conditions

The analysis of sufficient conditions is an analysis of whether the intersection of condition sets constitutes a subset of the result sets [55]. The sufficiency of the configuration is measured in terms of consistency and is calculated as follows:

$$Consistency(X_i \leq Y_i) = \sum(min(X_i, \ Y_i))/\sum(X_i) \tag{3}$$

A configuration is considered to be a subset of the result set when the consistency of the configuration is greater than the set threshold. At present, the consistency threshold for most studies is set to 0.75 or 0.8 [49]. The frequency threshold and PRI consistency threshold should be set to avoid conflicting configurations [55]. According to these requirements, the frequency threshold is 2, the consistency threshold is 0.8, and the PRI threshold is 0.75.

fsQCA 3.0 software is used to analyze the sufficiency of the configuration. This approach refers to the practices of Ragin and Fiss; thus, the output results are reported by listing the intermediate solutions supplemented by parsimonious solutions [49].

The membership scores of the cases covered by the path are shown in Fig 4.

As shown in Table 6 and Fig 4, the minimum values of consistency for the configurations are all higher than the minimum standard of 0.75 [55], and the solution consistency is 0.9188, which is also higher than 0.75; thus, the performance of the configurations and the solution is excellent.

The minimal formula of the outcome is shown in Eq (4):

$$(\sim TI* \sim IP*ME* \sim BE*RD*CD) + (TI*IP*ME* \sim BE*RD*CD) + (\sim TI*IP* \\ \sim ME*BE*RD*CD) + (\sim TI*IP*ME* \sim BE* \sim RD* \sim CD) + (TI* \sim IP* \sim ME* \\ \sim BE*RD* \sim CD) \rightarrow BI \tag{4}$$

In Configuration 1 (~TI*~IP*ME*~BE*RD*CD), the presence of cultural distance and the absence of a business environment are the core conditions, while the absence of technological innovation capability and intellectual property protection and the presence of market expansion capability and relational embeddedness are the peripheral conditions. Configuration 1 covers two cases. In Configuration 2 (TI*IP*ME*~BE*RD*CD), the core conditions are the presence of intellectual property protection and cultural distance and the absence of a business environment. The peripheral conditions are the presence of technological innovation capability, market expansion capability and relationship embeddedness. The number of cases covered by Configuration 2 is 2. The remaining configurations are explained in a similar manner.

**Table 5. Necessity analysis.**

| Condition | Brand internationalization | | Condition | Brand internationalization | |
|---|---|---|---|---|---|
| | Consistency | Coverage | | Consistency | Coverage |
| TI | 0.4713 | 0.7081 | BE | 0.5789 | 0.6763 |
| ~TI | 0.7523 | 0.6932 | ~BE | 0.6552 | 0.7321 |
| IP | 0.6312 | 0.6984 | RE | 0.5061 | 0.7894 |
| ~IP | 0.6015 | 0.7100 | ~RE | 0.6933 | 0.6247 |
| ME | 0.6546 | 0.7278 | CD | 0.6108 | 0.6808 |
| ~ME | 0.5464 | 0.6418 | ~CD | 0.6221 | 0.7287 |

Note: ~ indicates the absence of a condition

Accordingly, the situation within the configuration and the situation between configurations are complex and will be further analyzed in the "Theoretical Analysis" section.

### 4.3. Robustness test

In this paper, we analyzed robustness by independently changing the consistency threshold and frequency threshold [58]. [59] proposed criteria for the robustness of QCA, namely, the

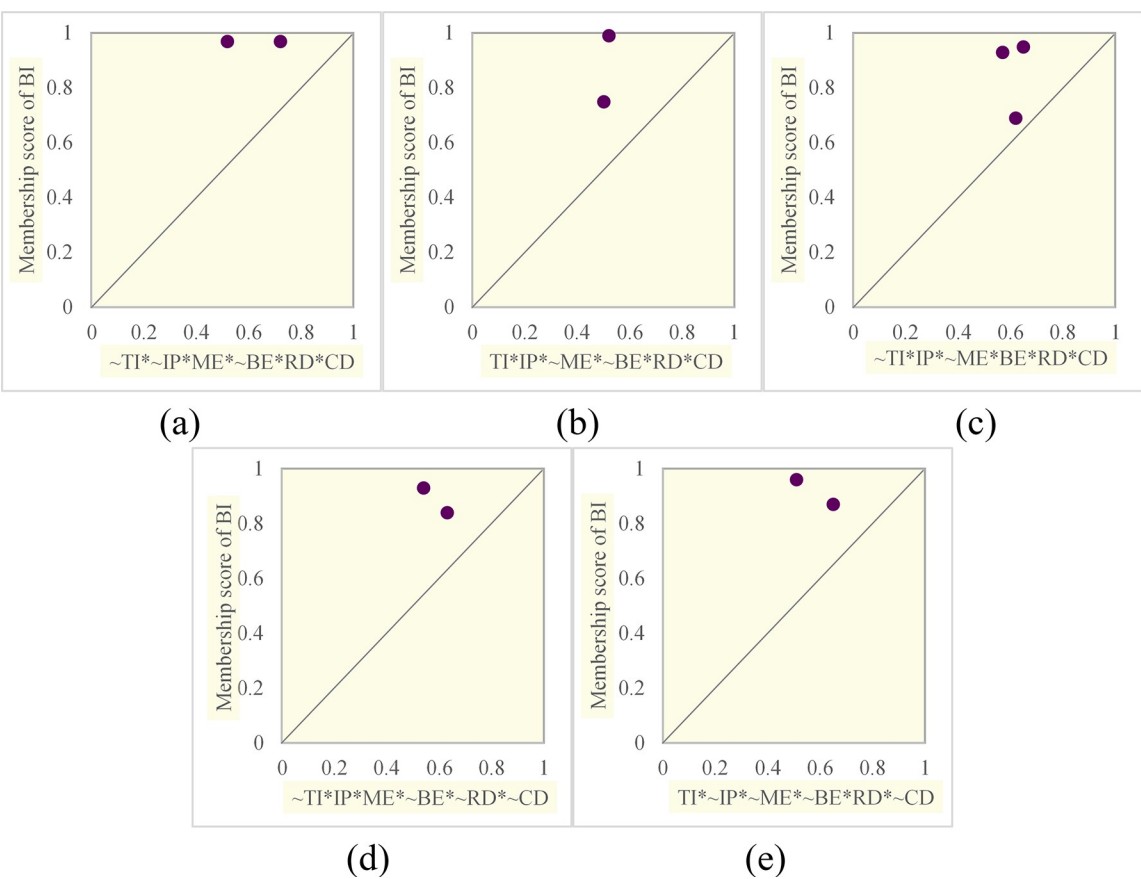

**Fig 4. Fuzzy-set plots for fsQCA solutions.** (a) Distribution of Configuration 1. (b) Distribution of Configuration 2. (c) Distribution of Configuration 3. (d) Distribution of Configuration 4. (e) Distribution of Configuration 5.

**Table 6. Configuration for brand internationalization.**

| Condition | Outcome: BI | | | | |
|---|---|---|---|---|---|
| | 1 | 2 | 3 | 4 | 5 |
| TI | ⊗ | ● | ⊗ | ⊗ | ● |
| IP | ⊗ | ● | ● | ● | ⊗ |
| ME | ● | ● | ⊗ | ● | ⊗ |
| BE | ⊗ | ⊗ | ● | ⊗ | ⊗ |
| RE | ● | ● | ● | ⊗ | ● |
| CD | ● | ● | ● | ⊗ | ⊗ |
| Row coverage | 0.1886 | 0.1568 | 0.2197 | 0.2095 | 0.1542 |
| Unique coverage | 0.0353 | 0.0300 | 0.0861 | 0.0808 | 0.0384 |
| Consistency | 0.9422 | 0.9893 | 0.9078 | 0.9776 | 0.9647 |
| Solution coverage | 0.4411 | | | | |
| Solution consistency | 0.9188 | | | | |

Note: ● (⊗) indicates the presence (absence) of a core condition; ● (⊗) indicates the presence (absence) of a peripheral condition.

parameters and set relationships of different configurations have not changed significantly. The results of the robustness test are shown in Table 7, referring to [60].

Table 7 shows that when the frequency threshold is reduced to 1 or the consistency threshold is reduced, the solution coverage is increased; thus, the new configuration is a superset of the original configuration. The set relationships among different configurations have not changed in general. The solution consistency is maintained at approximately 0.9, which is similar to the original solution consistency. In general, the robustness test results meet the criteria for the robustness of the QCA results.

## 5. Discussion

### 5.1. Theoretical analysis

According to the configuration table containing the output results, many ambiguities continue to require theoretical explanation. Therefore, the results must be further optimized and summarized based on the framework of brand internationalization to identify the path of Chinese brand internationalization.

**Table 7. Robustness test.**

| Threshold | 2/0.9078/0.75 | 1/0.9078/0.75 | 2/0.8771/0.68 |
|---|---|---|---|
| Configuration | Primitive solution | Variation in the frequency threshold | Variation in the consistency threshold |
| Configuration 1 | √ | [√] | √ |
| Configuration 2 | √ | [√] | √ |
| Configuration 3 | √ | √ | √ |
| Configuration 4 | √ | [√] | √ |
| Configuration 5 | √ | [√] | [√] |
| Solution Consistency | 0.9188 | 0.9266 | 0.8986 |
| Solution Coverage | 0.4411 | 0.5024 | 0.4804 |

Notes: Threshold: frequency/consistency/PRI; √ indicates that the configuration is the same as in the original configuration; [√] indicates that the configuration is a superset of the original configuration.

**Table 8. Categorization of paths.**

| Path | Brand empowerment | Boundaries spanned | Boundaries not spanned | Cases covered by path |
|---|---|---|---|---|
| Path 1 | Brand penetration; Brand acculturation | Spanning the market boundary under conditions of adversity; Spanning the cultural boundary under conditions of adversity | Adversity | Shein, Gearbest |
| Path 2 | Brand identity construction; Brand penetration; Brand acculturation | Spanning the technological boundary under conditions of prosperity; Spanning the market boundary under conditions of adversity; Spanning the cultural boundary under conditions of adversity | | Hisense, Huawei |
| Path 3 | Brand acculturation | Spanning the cultural boundary under conditions of adversity | Prosperity | Anker Innovations, China Eastern Airlines, ELEX |
| Path 4 | Brand penetration | Spanning the market boundary under conditions of adversity | Prosperity | YOOZOO, Tuya |
| Path 5 | Brand identity construction; Brand acculturation | Spanning the technological boundary under conditions of adversity; Spanning the cultural boundary under conditions of prosperity | Adversity | Xiaomi, Transsion |

According to the research framework for brand internationalization, the different capabilities of a brand are internal drivers of a brand's boundary-spanning ability, and they have a regulatory effect on the boundaries represented by external drivers [41]. Therefore, the presence of an internal driver indicates that the brand has the capability to span the corresponding boundary. The presence of an external driver indicates that the brand is facing a homeopathic environment. As cultural distance is a reverse variable, the absence of cultural distance indicates that the brand is facing a homeopathic environment. This paper classifies each path based on the brand's capabilities and the environment that the brand faces, as shown in Table 8.

**5.1.1. Analysis of cases covered by the paths.** As shown in Table 8, according to the outcome of fsQCA, the cases covered by Path 1 are Shein and Gearbest, which span market boundaries through excellent market expansion capability that empowers brands with penetration capability and which span cultural boundaries based on strong relational embeddedness with consumers that empowers brands with brand acculturation capabilities. Shein tracks global fashion trends using AI-derived big data technology, quickly entering more than 150 countries. Simultaneously, Shein focuses on improving user experience as its guiding goal by building a "brand platform". These factors empower brands with penetration and acculturation capabilities. Gearbest relies on the well-established overseas warehouse advantages of its parent company, Globalgrow e-commerce, to seize the overseas market quickly, sparing no efforts to establish itself on overseas social media platforms and collaborating with major Key opinion leaders (KOLs) to create high-quality content with the goal of enhancing the brand's penetration and acculturation capabilities.

The cases covered by Path 2 are Hisense and Huawei. Both enterprises have strong technological innovation capability, market expansion capability, and relational embeddedness capability that span technological, market, and cultural boundaries, thus empowering them with brand identity construction, penetration, and acculturation capabilities. Hisense and Huawei stand out in the global market due to their advantages with regard to providing high-quality products and being innovative, which empower these brands with brand identity construction capability. Extending into more than 160 countries and maintaining high interaction with consumers through increased brand exposure on social media, these enterprises thereby develop into global brands with strong comprehensive capabilities.

The cases covered by Path 3 are Anker Innovations, China Eastern Airlines and ELEX. All of these enterprises exhibit strong relational embeddedness with consumers, which can enable brands to span cultural boundaries and empower them with acculturation capability. Anker satisfies the unique needs of consumers due to its excellent appearance design, thereby

establishing strong embeddedness with consumers. As a service industry brand, China Eastern Airlines provides customer-oriented and cutting-edge services, thereby meeting the needs of customers from different cultural backgrounds and improving its brand acculturation capability. ELEX connects strangers from different cultural backgrounds by using social games as a medium and reduces cultural differences in the game context by establishing strong relationships with consumers, thereby enhancing the gaming experience of players from different cultural backgrounds.

The cases covered by Path 4 are YOOZOO and Tuya. Both YOOZOO and Tuya span market boundaries through their excellent market expansion capabilities, which can empower brands with penetration capability. YOOZOO and Tuya entered more than 200 countries over a period of 10 years and thus exhibited excellent market expansion capabilities, and their outstanding penetration capabilities drive these brands toward internationalization.

The cases covered by Path 5 are Xiaomi and Transsion. Due to their strong technological innovation capability and relational embeddedness capability, Xiaomi and Transsion span technological and cultural boundaries, thus empowering them with brand identity construction capability and acculturation capability. Due to its excellent community operation capability, Xiaomi maintains a high degree of embeddedness with consumers. Xiaomi focuses on a low-cost business model involving online channel sales and increases its investment in R&D with the goal of providing products that feature a high quality and low price, thus ensuring that Xiaomi is a preferred brand for consumers. Transsion launches a mobile phone that features a camera that is suitable for African consumers and that supports high-volume music playback. Transsion reaches deep into the hearts of consumers by exhibiting excellent and consistent product innovation.

**5.1.2. Comprehensive analysis of the paths.** By extracting common factors to organize Eq (4) [49], a structured formula can be obtained as follows in Eq (5):

$$ME*\sim BE*RD*CD*\begin{cases}\sim TI*\sim IP\\ TI*IP\end{cases}+(\sim TI*IP*\sim ME*BE*RD*CD)$$

$$+(\sim TI*IP*ME*\sim BE*\sim RD*\sim CD)$$

$$+(TI*\sim IP*\sim ME*\sim BE*RD*\sim CD)\rightarrow BI \qquad (5)$$

Eq (5) indicates that Paths 1 and 2 can be structured as

$ME*\sim BE*RD*CD*\begin{cases}\sim TI*\sim IP\\ TI*IP\end{cases}$, and it is evident that the common part of the two paths is

$ME^{*}\sim BE^{*}RD^{*}CD$. Based on the analysis shown in Table 8, we believe that in Paths 1 and 2, excellent market expansion capability enables brands to span market boundaries and then empowers them with penetration capability, whereas strong relational embeddedness capability enables brands to span cultural boundaries and empowers them with acculturation capability. That is, brand penetration capability and acculturation capability are the key capabilities for brands that focus on Paths 1 and 2 to achieve internationalization.

Combining Eq (5) and Table 8, we obtain the following findings:

1. Both paths that empower brands with a single capability and those that empower brands with multiple capabilities are evident. With regard to the richness of the capabilities with which the brand is empowered, Paths 3 and 4 are both single-capability paths that empower the brand with acculturation capability and penetration capability, respectively. Paths 1, 2

and 5 are multicapability paths, all of which empower the brand with acculturation capability.

2. The brand's penetrating capability and acculturation capability are the key capabilities for brand internationalization and have a certain substitution effect. With regard to the number of boundaries spanned, each path must span at least one boundary, but no path spans only the technological boundary. Spanning the market boundary and the cultural boundary are more important to brand internationalization than spanning the technological boundary, thus indicating that the brand's penetration capability and the brand's acculturation capability are more important and have a certain substitution effect.

3. A brand with a single capability is more likely to choose a positive environment. With regard to the characteristics of the boundaries not spanned, such boundaries are all positive in the single-capability paths. According to the eclectic theory of international production, a positive environment indicates that the host country offers a geographical advantage with respect to brand internationalization [61, 62]. Therefore, brands with a single ability often choose to enter a host country that features a positive environment corresponding to their lack of capabilities.

4. Paths 1, 2 and 3 are leapfrog internationalization paths, while Paths 4 and 5 are progressive paths. Four of the five paths span the cultural boundary, and most host countries that brands choose to enter are countries that feature a large cultural distance. According to springboard theory [57], the brand internationalization mode of emerging economies is different from the traditional progressive internationalization mode, and enterprises in such countries prefer to choose the leapfrog internationalization mode. The leapfrog internationalization mode emphasizes the fact that the brands of emerging economies first enter countries that are located far away from their home countries, such as in terms of large geographical or cultural distances [63], with the goal of obtaining the necessary resources and opportunities to reverse their own disadvantages and subsequently use those resources as a springboard for international expansion. Therefore, according to the presence of CD, three paths of leapfrog-type internationalization and two paths of progressive-type internationalization can be observed.

Accordingly, this paper selects three criteria to integrate the five brand internationalization paths: brand internationalization mode (leapfrog mode, progressive mode), brand empowerment richness (single-capability, multicapability) and type of boundary spanned (market-oriented, culture-oriented, comprehensive). Based on the richness of brand empowerment, this paper defines a path that empowers brands with a single capability as a single-capability path, while a path that empowers brands with more than two capabilities is defined as a multicapability path. With regard to the type of boundary spanned, this paper defines a path that features a spanned boundary that contains a market boundary (cultural boundary) as a market-oriented (culture-oriented) path and a path that contains both spanned boundaries as a comprehensive path. Four paths emerge following integration: Paths 1 and 2 are "leapfrog multicapability comprehensive" paths (L-M-Co), thus confirming Eq (5); Path 3 is the "leapfrog single-capability culture-oriented" path (L-S-Cu); Path 4 is the "progressive single-capability market-oriented" path (P-S-M); and Path 5 is the "progressive multicapability culture-oriented" path (P-M-Cu). The spaces and distributions of these paths are shown in Fig 5.

Fig 5 presents a total of 12 types of paths, which are divided according to the three dimensions. However, the facts indicate that there is no "single-capability comprehensive" path. Therefore, two types of paths are excluded. In fact, 10 kinds of paths are observed. The paths

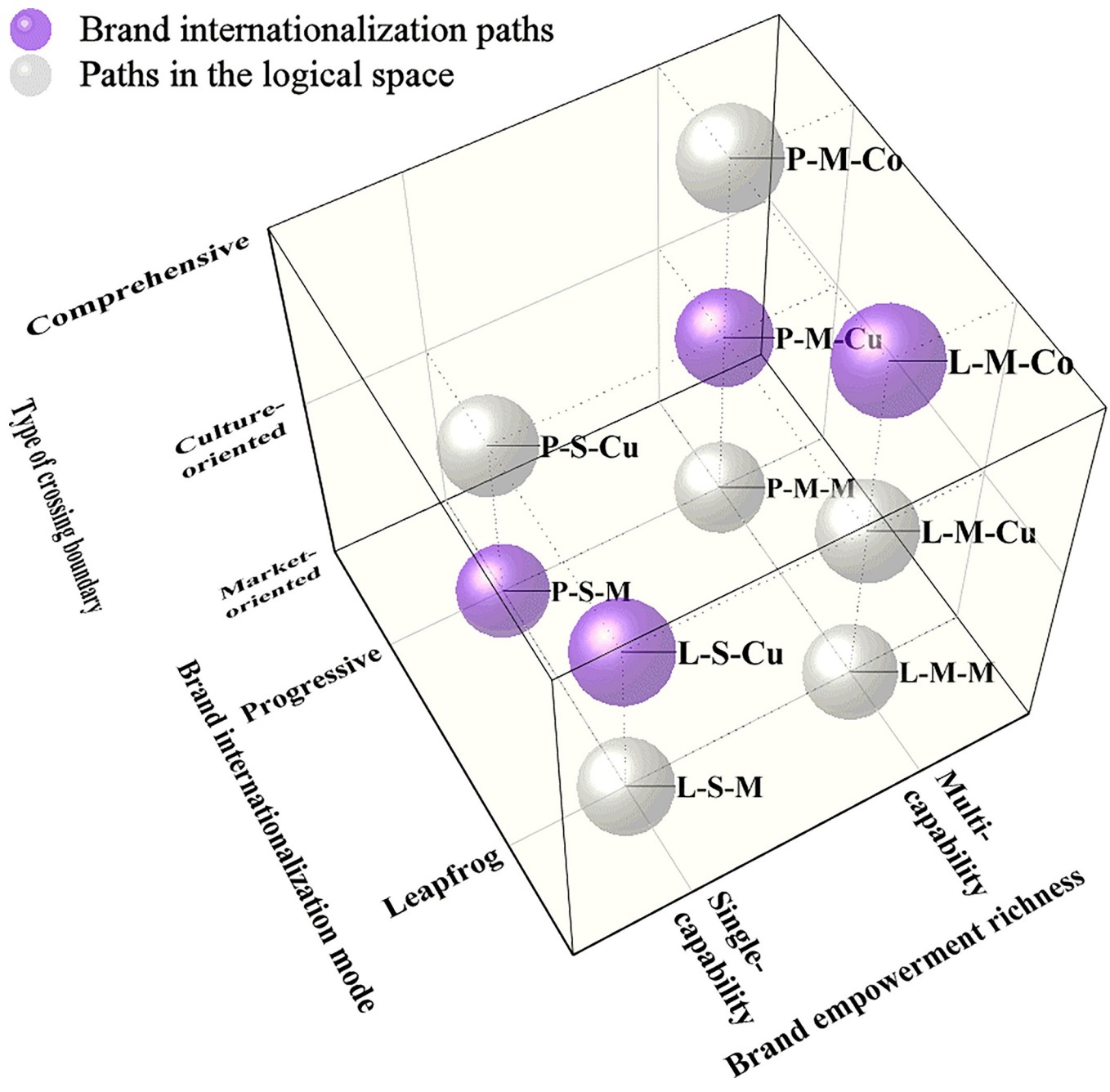

**Fig 5. 3D graph of the paths.**

revealed in this paper occupy 4 kinds of path spaces, as shown by the purple path in Fig 5. This logical space includes 6 paths, as shown in the gray path in Fig 5.

### 5.2. Case analysis

Based on the configuration table and the theoretical analysis, we obtain the four paths for brand internationalization, which are shown in Table 9; we then analyze the representative cases pertaining to each path to try to obtain practical support from the cases.

**Table 9. Brand internationalization paths.**

| Path | Cases covered by path | Representative case |
|---|---|---|
| L-M-Co | Shein, Gearbest, Hisense, Huawei | Huawei |
| L-S-Cu | Anker Innovations, China Eastern Airlines, ELEX | Anker Innovations |
| P-S-M | YOOZOO, Tuya | YOOZOO |
| P-M-Cu | Xiaomi, Transsion | Transsion |

The representative case of the "L-M-Co" path is Huawei. Although this study selected brands whose main business is B2C, Huawei engaged primarily in B2B business in the initial stage and began operating B2C mobile phone business in 2003. Huawei was founded in 1987 and began implementing its brand internationalization strategy in 1999. Within four years, it entered the United States (CD = 4.1) and Sweden (CD = 4.9), establishing R&D centers. Huawei, which began to operate its B2C consumer electronics business in 2003, entered markets such as the Netherlands (CD = 4.2) and the United Kingdom (CD = 3.5) by establishing joint ventures within three years. Accordingly, Huawei's brand internationalization in both a B2B and B2C context initially enabled the country to enter countries that are close to China in terms of cultural distance (i.e., greater than the intersection point, CD = 2.7), representing a typical leapfrog internationalization model. The establishment of international R&D centers and joint ventures with host country enterprises also confirm the springboard theory, which posits that brands first enter developed countries to seek resources and opportunities. In 2008, Huawei was ranked first worldwide in terms of its number of patent applications, and its excellent technological innovation capability empowered it with identity construction capability. In the same year, Huawei was ranked third in terms of market share in the field of mobile communications, and its excellent market expansion capability empowered it with brand penetration capability. Huawei established official Twitter social media accounts for the regions and countries that it had entered. The company's main account, @Huawei, has more than 8 million followers. Good relational embeddedness empowered the brand with acculturation capability. Based on the analysis of Huawei, Huawei's brand internationalization path is the "leapfrog multicapability comprehensive" path.

The representative case of the "L-S-Cu" path is Anker Innovations. Anker is a world-famous smart home brand that was founded in 2011. Initially, Anker established a goal of brand internationalization. It entered the markets of the US, the UK, Germany and other European and American countries in that same year. It entered Southeast Asia, the Middle East and other markets in 2017. This enterprise follows a typical leapfrog brand internationalization path. As a born global company, Anker also exhibits good relational embeddedness. Anker has established official accounts for each market segment; its employees are drawn from 30 countries, and the company has 15 offices worldwide. Anker's brand internationalization path is a "leapfrog single-capability culture-oriented" path with the advantage of brand acculturation capability.

The representative case of the "P-S-M" path is YOOZOO, which is a game brand. This enterprise was founded in 2009 and started brand internationalization in 2010. It first entered Singapore (CD = 0.6), Malaysia (CD = 1.2), the Philippines (CD = 1.3) and other developing countries that are close to China in terms of cultural distance (i.e., less than CD = 2.7). The brand internationalization mode of YOOZOO is progressive. In 2016, through the acquisition of Bigpoint, a well-known European game developer, YOOZOO entered the markets of developed countries such as Europe, the US and Japan. The brand internationalization of YOOZOO is a "progressive single-capability market-oriented" path dominated by brand penetration capability.

The representative case of the "P-M-Cu" path is Xiaomi, which was founded in 2010 and started the internationalization process in 2014. This enterprise first entered Southeast Asian countries, such as Singapore (0.6) and India (0.7), and subsequently entered European and American markets in 2017. Xiaomi's internationalization mode is the progressive internationalization mode. From 2020, Xiaomi has been listed among the "The Most Innovative Companies", published by BCG, for three consecutive years. Xiaomi has made remarkable achievements in innovation, and its excellent technological innovation capability empowers it with identity construction capability in the process of internationalization. Simultaneously, Xiaomi initially focused mainly on community operations to increase fan stickiness, and the company holds the "Mi Fan Festival" for its fans every year. During the 6th "Mi Fan Festival" in 2017, the number of online participants exceeded 57.4 million, indicating that Xiaomi maintains relatively close embedded relationships with its users and that the Xiaomi brand exhibits excellent brand acculturation capability. The analysis of Xiaomi reveals that Xiaomi's brand internationalization path is a "progressive multicapability culture-oriented" path.

The global smartphone industry entered its growth stage in approximately 2009 [64]. Huawei entered the international smartphone market in 2004, and Xiaomi started the internationalization process in 2014. Huawei started the internationalization process during the introduction stage of the industry, while Xiaomi did so in the growth stage. Anker Innovations entered the international smart home industry in 2011, when the smart home industry was in the introduction stage [65]. The electronic game industry entered its maturity stage in 1993 [66], and YOOZOO entered the game market in 2009. Thus, YOOZOO entered the international market during the maturity stage of the electronic games industry.

The case analysis of the leapfrog path reveals that the representative cases all entered the international market during the introduction stage of the industry and obtained resources and opportunities to empower these brands in the process of brand internationalization. This finding is also in line with industry life cycle theory. During the introduction stage, brands have fewer competitors. At this time, they can adopt the leapfrog internationalization strategy to gain a first-mover advantages and seize the major international market [67]. However, the case analysis of the progressive path reveals that the brands entered the international market in the growth and maturity stages. Therefore, the brand adopted the progressive internationalization strategy, entering neighboring markets and submarkets first and choosing to strengthen the brand's key penetration or acculturation capabilities to promote internationalization [68]. Combining the results of the theoretical analysis and case analysis, the four brand internationalization paths are all supported by these cases in practice.

## 6. Conclusions

Based on the research framework of brand internationalization empowerment, we use the fsQCA method and data drawn from 61 representative brands in China to explore the joint influences of internal and external drivers of brand internationalization, namely, technological innovation capability and intellectual property protection, market expansion capability and business environment, relational embeddedness and cultural distance, on brand internationalization. The conclusions of this paper are as follows:

### (1) Brand internationalization modes

Two modes of Chinese brand internationalization are evident: the progressive brand internationalization mode and the leapfrog brand internationalization mode. In addition to the progressive brand internationalization mode, which is similar to the mode employed in developed economies, the brand internationalization mode employed in China has also been found to be

**Table 10. Internationalization paths of Chinese brands.**

| Path | Initial stage of brand internationalization | In the process of brand internationalization | Remarks |
|---|---|---|---|
| L-M-Co | Quickly enter the world mainstream market (especially the developed market) | Continuous brand empowerment is implemented to facilitate the multidimensional and balanced development of the brand with regard to the capabilities of identity construction, penetration and acculturation. | |
| L-S-Cu | | Advantageous breakthrough development focusing on empowering brand acculturation capability | When choosing the single-capability path, brands should enter a positive environment. At this time, the host country has a location advantage, which is more beneficial to the internationalization and development of the brand. |
| P-S-M | Enter neighboring or similar markets first | Divergent development with a focus on empowering brand penetration capability | |
| P-M-Cu | | Focused development with an emphasis on empowering brand identity construction capability and brand acculturation capability | |

unique: the leapfrog brand internationalization mode. This finding also supports the claim that the internationalization path taken by brands in emerging economies is not a single model but rather exhibits diverse characteristics [4].

## (2) Brand internationalization paths

The internationalization paths of Chinese brands are mainly divided into four types, as shown in Table 10: the "leapfrog multicapability comprehensive" path (L-M-Co), the "leapfrog single-capability culture-oriented" path (L-S-Cu), the "progressive single-capability market-oriented" path (P-S-M), and the "progressive multicapability culture-oriented" path (P-M-Cu).

**(i) Differences from previous studies.** First, among the paths discussed in this article, two single-capability paths are included, namely, P-S-M and L-S-Cu. This fact stands in contrast to previous studies that have emphasized the need for comprehensive capabilities in multiple aspects for brand internationalization [6]. In the new industry context, we found that crossing a single boundary (market/culture) and possessing a single capability (penetration/acculturation) can also facilitate brand internationalization.

Second, this article classifies brand internationalization paths based on three dimensions: brand internationalization mode (leapfrog mode, progressive mode), brand empowerment richness (single-capability, multicapability) and type of boundary spanned (market-oriented, culture-oriented, comprehensive). It identifies six other logical paths that are not covered by the research sample on which this article focuses, thereby providing a reference and-direction for future research.

**(ii) Similarities to previous studies.** The two paths of L-M-Co and P-M-Cu revealed in this article support the previously made claim that brand internationalization typically requires multiple abilities, spanning multiple boundaries such as technology, market, and culture [1, 6, 57].

## (3) Brand internationalization capabilities

Brand penetration and acculturation are key capabilities with regard to brand internationalization and have a certain substitution effect. Contrary to the conclusions of previous research on technology-oriented and market-oriented brand internationalization paths [40], we find that the paths of brand internationalization can be divided into two categories, i.e., market-oriented and culture-oriented, indicating that brand penetration capability and brand acculturation capability are the key capabilities pertaining to brand internationalization and have a substitution effect.

## 7. Implications

### 7.1. Managerial implications

Based on the conclusions discussed above, the research can provide the following practical insights for enterprises in China and other emerging economies:

1. First, the mode of internationalization of the brand should be chosen in accordance with the stage in the life cycle to which the industry in which the brand is located has progressed. If the industry in which the brand is located is in the introduction stage, the brand can adopt the leapfrog brand internationalization mode to seize the available market opportunity. Second, in accordance with their own resources and capabilities, brands should choose whether to take the "L-M-Co" path or the "L-S-Cu" path. Regardless of which path these brands choose, they should pay attention to brand empowerment in terms of penetration capability and acculturation capability throughout the process of internationalization. When brands belong to the service industry, they should try to choose the leapfrog single-capability path of L-S-Cu but also pay attention to crossing cultural boundaries; in addition, they should focus on empowering brand acculturation capability and choose a host country that features a good business environment and strong intellectual property protection.

2. If the industry in which the brand is located is in the growth or maturity stage, the brand can employ the progressive mode of brand internationalization and subsequently choose a brand internationalization path based on its own capabilities and brand internationalization strategy. If a divergent market strategy is employed, brands should give priority to the "P-S-M" path. At this time, attention should be given to the importance of choosing a host country that features a high degree of intellectual property protection. If a focused cultural strategy is employed, brands should prioritize the "P-M-Cu" path.

### 7.2. Theoretical implications

The main contributions of this paper are as follows:

1. This paper broadens the theoretical boundaries of brand internationalization capability. Although previous research has considered the influence of internal and external factors in this context, research on the interaction and joint effects among multiple factors on brand internationalization remains lacking [8]. Brand penetration and acculturation are key capabilities with respect to brand internationalization and have a certain substitution effect with regard to fsQCA, thus broadening the theoretical boundaries of brand internationalization capability.

2. The paper also advances the research on empowering brand internationalization. The process of enterprise internationalization is also a process of continuously empowering brands with internationalization capabilities [6], and the mechanism underlying brand empowerment in the process of enterprise internationalization is not yet clear [1, 6]. This article combines brand empowerment theory and boundary spanning theory to propose a brand empowerment framework for brand internationalization. The relationship between brand empowerment and brand boundary spanning is elucidated by reference to the mechanism of brand internationalization empowerment. In addition, we propose research ideas for empowering brand internationalization.

3. This study enriches the research on the internationalization of Chinese brands in terms of the sample under investigation. Previous research has focused mostly on the manufacturing industry [3], but the internationalization path of brands exhibits diversity, which is

influenced by industry heterogeneity. The diversity of the internationalization path of Chinese brands also requires further exploration [4]. This article is based on 61 representative Chinese brands spanning multiple segmented industries and organizes the sample data in terms of six dimensions, such as technological innovation capability and market expansion capability. This approach enriches the research on Chinese brand internationalization in terms of the sample under investigation.

4. This study proposes four paths for the internationalization of Chinese brands and reveals six logical paths. Most previous research on brand internationalization in China has focused on the single stage of brand internationalization, and such research has rarely considered the multiple stages of brand internationalization [69]. Based on the research framework of brand empowerment, this article proposes four paths for Chinese brand internationalization based on fsQCA and theoretical analysis, thereby providing an internationalization path reference for other brands. Simultaneously, two new paths are discovered, thus suggesting new alternative paths for Chinese brands' internationalization. We also reveal six logical paths for brand internationalization that can be explored in future research.

## 8. Limitations and directions for future research

### Limitations

Although the aim of this research was to be as perfect as possible, it nevertheless has certain limitations.

1. Due to limitations pertaining to data availability, the data referenced in this paper were selected in 2019, which may have caused problems as a result of an insufficient time span.

2. Six dimensions were selected, and some dimensions may be missing.

3. Because the standardized process of the fsQCA method requires the selection of the antecedent variable based on the extant literature and the researchers' theoretical basis, this approach limits the thinking of researchers to some extent [70].

### Directions for future research

One possible direction for the expansion of related research is as follows:

1. Brand internationalization can be divided based on to three standards, and there are 10 paths in logical space; however, the results of this paper cover only 4 paths, thus leaving open the possibility of exploring the remaining paths.

2. Due to the continuous development of digital technology, the convenience of digital technology has created new possibilities for brand internationalization. It is necessary to consider the impact of digital technology on brand internationalization in subsequent research [71], a task which our future research addresses.

## Supporting information

**S1 Data.**
(XLSX)

## Author Contributions

**Conceptualization:** Junfeng Liao, Minru Yang.

**Data curation:** Junfeng Liao, Minru Yang.

**Formal analysis:** Junfeng Liao, Minru Yang.

**Funding acquisition:** Junfeng Liao.

**Investigation:** Minru Yang.

**Methodology:** Junfeng Liao.

**Project administration:** Junfeng Liao.

**Resources:** Junfeng Liao.

**Software:** Minru Yang.

**Supervision:** Junfeng Liao.

**Validation:** Junfeng Liao, Minru Yang.

**Visualization:** Minru Yang.

**Writing – original draft:** Minru Yang.

**Writing – review & editing:** Junfeng Liao.

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
