## [Decision Letter · Decision Letter 0]

6 Feb 2023

PONE-D-22-33203How to Travel on the Road of Internationalization of Emerging Economy Brands: Evidence from Representative Brands in ChinaPLOS ONE

Dear Dr. Liao,

Thank you for submitting your manuscript to PLOS ONE. After careful consideration, we feel that it has merit but does not fully meet PLOS ONE’s publication criteria as it currently stands. In view of the referees’ feedback and my own reading of your paper, we invite you to address all issues noted below. We consider these issues to be major in nature, requiring more than a superficial or minor revision. In particular, one of the reviewer find that the contribution to the literature on this topic is quite limited on a very studied topic as this one. The reviewer also suggests paying more attention to the literature review section.Therefore, we invite you to submit a revised version of the manuscript that addresses the points raised during the review process.

 Please submit your revised manuscript by Mar 23 2023 11:59PM. If you will need more time than this to complete your revisions, please reply to this message or contact the journal office at plosone@plos.org. Please include the following items when submitting your revised manuscript:A rebuttal letter that responds to each point raised by the academic editor and reviewer(s). You should upload this letter as a separate file labeled 'Response to Reviewers'.A marked-up copy of your manuscript that highlights changes made to the original version. You should upload this as a separate file labeled 'Revised Manuscript with Track Changes'.An unmarked version of your revised paper without tracked changes. You should upload this as a separate file labeled 'Manuscript'.

We look forward to receiving your revised manuscript.

Kind regards,

J E. Trinidad Segovia

Section Editor

PLOS ONE

Journal Requirements:

Reviewers' comments:

Reviewer's Responses to Questions

**Comments to the Author**

1. Is the manuscript technically sound, and do the data support the conclusions?

Reviewer #1: Yes

Reviewer #2: Partly

2. Has the statistical analysis been performed appropriately and rigorously? 

Reviewer #1: I Don't Know

Reviewer #2: N/A

3. Have the authors made all data underlying the findings in their manuscript fully available?

Reviewer #1: Yes

Reviewer #2: No

4. Is the manuscript presented in an intelligible fashion and written in standard English?

Reviewer #1: Yes

Reviewer #2: Yes

5. Review Comments to the Author

Reviewer #1: This article addresses the still new but growing theme of internationalization of brands from emerging economies. While limited in its scope and number of cases, this article presents a creative way of looking at our current understanding of the factors and indicators affecting brand infiltration, brand acculturation and ultimately, the ability to succeed internationally. This study may be useful, going forward, to larger, more comprehensive and comparative studies involving more than one country and greater number of brands.

Reviewer #2: This manuscript focus on the internationalization of Chinese brands. This topic is worthy to study. The authors have made limited contributions to Chinese areas. The authors should pay more energy to improve this manuscript

6. PLOS authors have the option to publish the peer review history of their article (what does this mean?). If published, this will include your full peer review and any attached files.

Reviewer #1: No

Reviewer #2: No

---

## [Author Response · Author response to Decision Letter 0]

30 Apr 2023

Dear Editors and Reviewers,

Thank you for your letter and for the reviewers' comments concerning our manuscript entitled "How to Travel on the Road of Internationalization of Chinese Brands" with Manuscript Number: PONE-D-22-33203. Those comments are all valuable and very helpful for revising and improving our paper, as well as the important guiding significance to our research. We have studied the comments carefully and have made corrections which we hope meet with approval. The revised portions are highlighted in grey in the paper. The main corrections in the paper and the responses to the reviewers' comments are as follows.

1. Response to comment: ("The title should be direct to reveal the internationalization of Chinese brands. The whole paper should clearly pay attention to this research object too. As second largest economy, Chinese brands have enough potential to be studied. The authors can write some words about implication for emerging economy in the end.")

Response: Thanks to the reviewers for pointing out this problem. Our previous perspective is indeed too large compared with our research samples that are all Chinese brands. We fully accept the suggestions. We have made some adjustments as follow.

(i) ("The title should be direct to reveal the internationalization of Chinese brands. "): We have modified the title to: "How to Travel on the Road of Internationalization of Chinese Brands" (lines 1)，

(ii) ("The whole paper should clearly pay attention to this research object too."): we have added more literatures on the internationalization of Chinese brands in the Introduction section (lines 21-86). Relevant details have also been adjusted (lines 11, 17, 18, etc.).

(iii) ("The authors can write some words about implication for emerging economy in the end."): we have written some words about implication for emerging economies in subsection 5.3. Practical Enlightenment (lines 572-574).

The relevant modifications are in section 1. Introduction and subsection 5.3. of the manuscript.

2. Response to comment: ("The authors should integrate more literature about internationalization Chinese companies. “Research gap” in the part “Introduction” should be in the last paragraph of “Introduction”. There are too many theoretical perspectives in “Introduction” and in the whole paper. There are too many research questions. The authors may highlight 2 or three questions.")

Response: Thanks to the reviewers for pointing out this problem. Our previous literature integration is indeed fragmented. According to the reviewers' suggestions, we have focused on the internationalization of Chinese brands and adjusted the research gap to the end of the Introduction section, and reduced the number of research questions from 4 to 2 to make our research more targeted.

(i) ("The authors should integrate more literature about internationalization Chinese companies."): we have organized the literature on brand internationalization of Chinese enterprises form three perspectives: “go global”, “be local” and “lead global” (lines 31-81).

(ii) ("“Research gap” in the part “Introduction” should be in the last paragraph of “Introduction”."): the research gap has been adjusted to the end of “Introduction” (lines 81-86).

(iii) ("There are too many theoretical perspectives in “Introduction” and in the whole paper."): we have focused on the internationalization path of Chinese brands (lines 68-97), and our research theory reduced to two theories: brand empowerment and boundary spanning (lines 98-142).

(iv) ("There are too many research questions. The authors may highlight 2 or three questions."): our research questions have also been refined to: (1) What are the key drivers of Chinese brands to achieve brand internationalization; (2) What are the paths for the Chinese brand internationalization (lines 92-94).

The relevant modifications are in section 1. Introduction and section 2. Research Model of the manuscript.

3. Response to comment: ("As above, there too many theories, for example, brand equity, brand identity, spring board, born global, RBV. “Cross the Chasm” has a special context. There are many literatures about “Liability of origin, liability of foreignness”. The authors may select one or two theories, and conduct literature review in detail, find research gap.")

Response: Thank you for pointing out that our theoretical perspective is quite scattered and the structure is not clear enough. We have integrated the literature review into the Introduction section to make the structure of the article clearer and more compact.

(i) ("As above, there too many theories, for example, brand equity, brand identity, spring board, born global, RBV."): There are indeed too many theories in this article, so we have narrowed them down and introduced the brand empowerment theory and the boundary spanning theory in detail (lines 98-142).

(ii) ("“Cross the Chasm” has a special context."): We are very sorry for our negligence. Cross the Chasm has a special context that is applicable to consumers' acceptance of new products, not brand internationalization. We have modified our main theory to boundary spanning theory (lines 98-142).

(iii) ("The authors may select one or two theories, and conduct literature review in detail, find research gap."): We have conducted the literature review as suggested and found research gap: “This means that there is a certain research gap in the multi-stage brand internationalization path research based on quantitative data and considering the interactive effect of multiple drivers”. Due to the theoretical support required for the proposed model of QCA research is very strict, we have presented a detailed explanation of the proposed research model as a chapter to reflect the exact of our model (section 2. Research Model, lines 98-167), and incorporated the literature review section into the Introduction section (lines 35-86). It is hoped that editors and reviewers can allow us to make such adjustments.

The relevant modifications are in section 1. Introduction and section 2. Research Model of the manuscript.

4. Response to comment: ("Table 6: The brands in Table 6 come from different industries, the companies have different histories. For example, in the beginning Huawei supplied only telecommunication equipment, namely B to B. Hisense mainly B to C. Some brands are platform. Xiaomi has completely different business model. Therefore, the authors should be aware of these differences as the authors discuss about paths of brand internationalization.")

Response: Thanks to the reviewers for pointing out this problem. We do not consider that readers' different understanding of the enterprise that makes our result confusing. Therefore, based on the suggestions of reviewers, we have added 4.4.1. subsection to provide a detailed explanation of the background of the cases. 

(i) ("The brands in Table 6 come from different industries, the companies have different histories "): The results shown in Table 7 (Table 6 of the original manuscript) are a little confusing. In order to provide readers with a clear understanding of the background of cases covered by path in the output result of QCA, we have added section 4.4.1. Analysis of cases covered by path (lines 303-351). Based on the theoretical framework of this article, we have provided a background introduction to the cases covered by each path (lines 303-351).

(ii) ("in the beginning Huawei supplied only telecommunications equipment, named B to B. "): we have introduced it in section 4.5. that Huawei's development process after starting B to C is in line with the L-M-Co path (lines 425-437).

(iii) In order to demonstrate the reason for the merging of Path 1 and Path 2, we have added two minimal formulas to reflect the similarity of the two paths by extracting common factors from the formulas for Path 1 and Path 2 (lines 260-263, 353-358). 

The relevant modifications are in subsection 4.2, 4.4 and 4.5 of the manuscript.

5. Response to comment: ("5.2 Research Contribution: The authors may conduct literature review, find research gap, then rewrite theoretical contributions.")

Response: Thanks to the reviewers for pointing out this problem. We did not integrate the literature review, research gap, and theoretical contributions well before. Therefore, we have conducted literature review (lines 35-87), found research gap (lines 82-87). Based on the research gap, we have rewritten theoretical contributions (lines 542-572).

The relevant modifications are in subsection 5.2. Theoretical Contributions of the manuscript.

6. Response to comment: ("Reference: The literature in the reference should have unified format.")

Response: Thanks to the reviewers for pointing out this problem. We have carefully examined each literature and found that there is indeed missing page section in some literatures. We have corrected the malformed citation (lines 602-743).

The relevant modifications are in section References of the manuscript.

We have tried our best to improve the manuscript and made some changes. The revised portions are highlighted in grey in the paper.

We appreciate the Editors' warm work and the Reviewers' valuable comments earnestly and hope that the correction will meet with approval.

Once again, thank you very much for your comments and suggestions.

---

## [Decision Letter · Decision Letter 1]

3 Jul 2023

PONE-D-22-33203R1How to Travel on the Road of Internationalization of Chinese BrandsPLOS ONE

Dear Dr. Liao,

Thank you for submitting your manuscript to PLOS ONE. After careful consideration, we feel that it has merit but does not fully meet PLOS ONE’s publication criteria as it currently stands. Therefore, we invite you to submit a revised version of the manuscript that addresses the points raised during the review process.

After the previous review rounds, one of the reviewers continues to say that the paper has deficiencies in terms of its contribution to the financial literature, which is why I decided to incorporate a third reviewer to make a final decision on it. The suggestions of the same and the previous one can be seen below. Both think that the manuscript requires important changes to be considered.

We look forward to receiving your revised manuscript.

Kind regards,

J E. Trinidad Segovia

Section Editor

PLOS ONE

Reviewers' comments:

Reviewer's Responses to Questions

**Comments to the Author**

1. If the authors have adequately addressed your comments raised in a previous round of review and you feel that this manuscript is now acceptable for publication, you may indicate that here to bypass the “Comments to the Author” section, enter your conflict of interest statement in the “Confidential to Editor” section, and submit your "Accept" recommendation.

Reviewer #2: All comments have been addressed

Reviewer #3: (No Response)

2. Is the manuscript technically sound, and do the data support the conclusions?

Reviewer #2: No

Reviewer #3: (No Response)

3. Has the statistical analysis been performed appropriately and rigorously? 

Reviewer #2: Yes

Reviewer #3: (No Response)

4. Have the authors made all data underlying the findings in their manuscript fully available?

Reviewer #2: Yes

Reviewer #3: (No Response)

5. Is the manuscript presented in an intelligible fashion and written in standard English?

Reviewer #2: No

Reviewer #3: (No Response)

6. Review Comments to the Author

Reviewer #2: Title

The title is too big and stays on the surface. The title should be clear, concrete, reflect the main contribution, or key point of this manuscritp，and therefore, attractive.

Abstract:

Maybe delete the two sentences in the beginning about “emerging economies”. Single case stduy should not be a “gap” or weak point. You cannot say mutlple case study is better than single case study.

Keywords:

fsQCA should not be a keyword.

Introduction

Please check and improve Table 1

At the same time, most of the existing brand internationalization path studies are case studies, and the sample size of the study is too small, so the conclusions obtained are statistically unreliable and need to be tested by quantitative research (Wei et al., 2015)..

This argument about case study is not correct.

Introduction in this manuscript should be divided into introduction and literature review.

The research gap focuses on methodology is not persuasive

2. Research model should be literature review. In literature review you should find a research gap.

Contribution

It has enriched the research samples of brand internationalization of China. Most

of the existing brand internationalization path studies are case studies, and the sample size

of the study is too small (Wei et al., 2015). Based on collecting data from 61 representative

Chinese brands, this article collates sample data in seven dimensions, including technological innovation capability, market expansion capability, etc. Enriching the

research samples of Chinese brand internationalization.

This paragraph is not correct. You may read some article about case study.

Reviewer #3: The reviewer believes that the topic “How to Travel on the Road of Internationalization of Chinese Brands” is worthy of investigation. However, the following needs to be addressed. There are minor and major issues that should be corrected. I believe the paper could be further strengthened by added information about.

Please revise the paper title according to the content.

Please reorganize the manuscript at the journal request. Please change the reference format.

The language of this manuscript is very bad and needs help from native speakers.

The title of the manuscript should fully demonstrate the content of this study and the relevant subjects.

Abstracts should include the purpose and findings of the study.

LINES28-34 This a very vague statement. These sentences do not provide any information on how the concept could be conceptualized?

LINES69-87. This section should explain the study's context and research objective. Furthermore, the research gap needs to be narrowed after analyzing the previous studies. The research method is not adequately explained in the first section.

-Introduction, what authors wanted to convey. Here author must build research gap following the previous studies.-The manuscript does not answer the following concerns: Why is it timeliness to explore such a study? What makes this study different from the previously published studies? Are there any similarly findings in line with the previously published studies? Are the findings different from prior academic studies that were conducted elsewhere, if any? For example, Transformation Drive Internationalization, what it requires, what are the new technologies, some recent issue highlights the importance. See the following: New Energy-Driven Construction Industry: Digital Green Innovation Investment Project Selection of Photovoltaic Building Materials Enterprises Using an Integrated Fuzzy Decision Approach. Systems 2023, 11, 11. https://doi.org/10.3390/systems11010011

-Methodology: Model.. I suggest authors here build your main heading on Research and data methodology. Clearly explain the model building process, and what previous studies have used similar models (model testing approach).

There is no flow in the text. It partly depends on the lack of proofreading but also on the fact that many statements and claims are made without being followed up by a clear and logical discussion. It is especially problematic in the Introduction that brings up a number of findings from different areas without linking them together.

Please make sure your conclusions' section underscores the scientific value-added of your paper, and/or the applicability of your findings/results. Highlight the novelty of your study.

In addition to summarizing the actions taken and results, please strengthen the explanation of their significance. It is recommended to use quantitative reasoning comparing with appropriate benchmarks, especially those stemming from previous work. See the following: An adoption-implementation framework of digital green knowledge to improve the performance of digital green innovation practices for industry 5.0, https://doi.org/10.1016/j.jclepro.2022.132608.

More importantly, the choice of the variables should be explained in light of the theory and the prior literature on the topic. The arguments are simply relationships and causes very close to the replication of many studies dealing with the same thing.

The authors should emphasize the important role of digital technology in future research. See the following: Enhancing Digital Innovation for the Sustainable Transformation of Manufacturing Industry: A Pressure-State-Response System Framework to Perceptions of Digital Green Innovation and Its Performance for Green and Intelligent Manufacturing. https://doi.org/10.3390/systems10030072

Please consider this structure for manuscript final part.

-Discussion

-Conclusion

-Managerial Implication

-Practical/Social Implications

-Discussion needs to be a coherent and cohesive set of arguments that take us beyond this study in particular, and help us see the relevance of what authors have proposed. Authors should create an independent “Discussion” section. Author need to contextualize the findings in the literature, and need to be explicit about the added value of your study towards that literature. Also other studies should be cited to increase the theoretical background of each of the method used. Findings should be contextualized in the literature and should be explicit about the added value of the study towards the literature. Limitations and future research.

As any emprical study that use different approaches I would like to ask to introduce in the Conclusion section at least a paragraph containing the study limitations. I noticed some things in the paper but a synthesis of statements related to how the study is useful (or partially useful, since are required certain further analysis) and helps potential interested readers does not really exist. Maybe in addition to the last section of Conclusion it is beneficial to introduce a section called: Discussion.

7. PLOS authors have the option to publish the peer review history of their article (what does this mean?). If published, this will include your full peer review and any attached files.

Reviewer #2: No

Reviewer #3: No

---

## [Author Response · Author response to Decision Letter 1]

2 Sep 2023

Dear Editors and Reviewers,

Thank you for your letter and for the reviewers' comments concerning our manuscript entitled " New Internationalization Paths of Chinese Brands: A Configurational Study" with Manuscript Number: PONE-D-22-33203R1. Those comments are all valuable and very helpful for revising and improving our paper, as well as the important guiding significance to our research. We have studied the comments carefully and have made corrections which we hope meet with approval. The revised portions are highlighted in grey in the paper. The main corrections in the paper and the responses to the reviewers' comments are as follows.

Reviewer #2:

1. Response to comment: ("The title is too big and stays on the surface. The title should be clear, concrete, reflect the main contribution, or key point of this manuscript, and therefore, attractive.")

Response: Thanks to the reviewers for pointing out this problem. As you mentioned, the title should concretely reflect our main contributions. Our previous title did not adequately do so. Regarding this issue, we have modified our title to "New Internationalization Paths of Chinese Brands: A Configurational Study" (lines 1-2) based on our key contributions of (1) synthesizing brand samples across multiple industries and discovering two new paths different from prior research, and (2) examining the complex causality of brand internationalization using fsQCA which revealed interactions among drivers and uncovered market-oriented and culture-oriented patterns previously undiscovered.

2. Response to comment: ("Abstract: Maybe delete the two sentences in the beginning about “emerging economies”. Single case study should not be a “gap” or weak point. You cannot say mutlple case study is better than single case study.")

Response: We are very grateful for the detailed and insightful feedback provided by the reviewer on our Abstract. It is clear that our previous abstract lacked proper organization, and we have rewritten it carefully following the suggestions.

("Abstract: Maybe delete the two sentences in the beginning about “emerging economies"): In revising the abstract based on feedback, we agreed removing the initial statements about emerging economies was appropriate, as the focus of our study is the current situation of Chinese brand internationalization. At the same time, we have supplemented the theoretical background concerning the research status of Chinese brand internationalization, as the introduction should set the context before describing the contributions (lines 8-15).

(Single case study should not be a “gap” or weak point. You cannot say mutlple case study is better than single case study.): As mentioned above, case study should not become a research gap. This is due to our previous lack of understanding of the case study. Through careful reading of literature and an in-depth understanding of case study, we believe that the biggest advantage of case analysis is that by analyzing one or more cases in detail, it can output a highly summarized conclusion or framework, To provide a very solid research foundation for subsequent research, the model and framework of our manuscript also benefit greatly from this. So we have removed the incorrect description of the case analysis in this manuscript and added section 2 The Literature Review section (lines 59-158) identified our research gap based on the Literature Review (lines 128-158).

The relevant modifications are in Abstract and section 2. Literature Review of the manuscript.

3. Response to comment: ("Keywords: fsQCA should not be a keyword.")

Response: Thank you very much for pointing out the non-standard aspects of our keywords in such detail. Keywords should reflect the research content of the literature, and fsQCA as a research method should not be included in the keywords section. Therefore, we have deleted the keyword fsQCA and re-selected the following keywords: Internationalization path; Brand empowerment; Chinese brands (line 24).

The relevant modifications are in Keywords of the manuscript.

4. Response to comment: ("Introduction: Please check and improve Table 1. At the same time, most of the existing brand internationalization path studies are case studies, and the sample size of the study is too small, so the conclusions obtained are statistically unreliable and need to be tested by quantitative research (Wei et al., 2015). This argument about case study is not correct. Introduction in this manuscript should be divided into introduction and literature review. The research gap focuses on methodology is not persuasive")

Response: Thank you very much for your question regarding the Introduction section of this manuscript. Your constructive suggestions have made us aware of the issues of unclear structure and inaccurate expression in the Introduction section, which is very beneficial for us to further improve our manuscript. In response to the above suggestions, we have made the following modifications.

("Introduction: Please check and improve Table 1."): Table 1 of the original manuscript was placed in section 1. Introduction, but as a summary table of relevant literature, we realized that this table should be placed in the 2. Literature Review section. Therefore, Table 1 of the original manuscript has become Table 2 of 2. Literature Review. At the same time, we also realized that Table 1 of the original manuscript provided a rough summary of the literature, so we further improved Table 2 (Table 1 of the original manuscript) based on the research problems, research subjects, research methods, and theoretical support (lines 70-71).

("At the same time, most of the existing brand internationalization path studies are case studies, and the sample size of the study is too small, so the conclusions obtained are statistically unreliable and need to be tested by quantitative research (Wei et al., 2015). This argument about case study is not correct."): We strongly agree with this suggestion, which is due to our previous lack of understanding of case study. Through careful reading of literature and in-depth understanding of case study, we believe that one of the biggest advantages of case analysis is that by analyzing one or more cases in detail, it can output a highly summarized conclusion or framework, providing a very solid research foundation for subsequent research, The model and framework of our manuscript also greatly benefit from this. So we have removed the incorrect description of the case analysis in this manuscript.

("Introduction in this manuscript should be divided into introduction and literature review. "): This suggestion is very useful for us. Introduction of the previous manuscript mixed the content of the introduction and literature review, resulting in a relatively rough statement in this section, which is not conducive to readers' better reading of our manuscript. So we have refined the Introduction section (lines 25-58) based on practical background, theoretical background, research gaps, and research content, and added the Literature Review section (lines 59-158) to obtain our research gap (lines 128-158).

(“The research gap focuses on methodology is not persuasive”): Thank you very much for pointing out this problem. This plays a very important role in improving our manuscript. We did not carefully review the existing literature to identify our research gaps, but instead focused on research methods, which is inappropriate. This is also a major issue in our previous version of the manuscript. Therefore, we have added 2. Literature Review section (lines 59-158) and organized relevant literature through Table 2, analyzing the research problems, research subjects, research methods and theoretical support (lines 70-71), and then to get our research gap (lines 128-158).

5. Response to comment: ("Research model should be literature review. In literature review you should find a research gap.")

Response: Thank you very much for pointing out the structural arrangement issue in our original manuscript. Our new manuscript focuses on adding 2. Literature Review section (lines 59-158) and organizing relevant literature through Table 2. We analyzed the research problems, research subjects, research methods and theoretical support (lines 70-71), to identify our research gap (lines 128-158). At the same time, we also placed the research model (lines 160-247) in 3. Research and Data Methodology section makes the structure of the manuscript smoother.

The relevant modifications are in section 1. Introduction, section 2. Literature Review and subsection 3.1. Research Model of the manuscript.

6. Response to comment: ("Contribution: It has enriched the research samples of brand internationalization of China. Most of the existing brand internationalization path studies are case studies, and the sample size of the study is too small (Wei et al., 2015). Based on collecting data from 61 representative Chinese brands, this article collates sample data in seven dimensions, including technological innovation capability, market expansion capability, etc. Enriching there search samples of Chinese brand internationalization. This paragraph is not correct. You may read some article about case study.")

Response: Thank you very much for pointing out the issue of unclear organization in our Contribution section. This suggestion is very helpful to us, and we have carefully revised our manuscript based on it. The main reason for the unclear expression of research contributions is that the original manuscript did not organize the Literature Review well. Therefore, we have added the 2. Literature Review section and rewritten our Contributions section based on the research gap (lines 669-729). In addition, in response to the incorrect description of case study in our manuscript, we have also deeply realized our lack of rigorous language organization in this section. Through careful reading of the literature and an in-depth understanding of case study, we believe that one of the biggest advantages of case study is that can output a highly summarized conclusion or framework by analyzing one or more cases in detail, which provides a very solid research foundation for subsequent research. Our manuscript's model and framework also benefited greatly from this. Therefore, we removed the erroneous description of case study in this manuscript and reproduced the contribution of “enriching the research sample”, mainly focusing on the aspect that our manuscript sample includes brands from multiple sub-industries (lines 711-719).

The relevant modifications are in subsection 7.2. Theoretical Implications of the manuscript.

Reviewer #3:

1. Response to comment: ("Please revise the paper title according to the content. The title of the manuscript should fully demonstrate the content of this study and the relevant subjects. ")

Response: Thanks to the reviewers for pointing out this problem. the title should concretely reflect our main contributions. Our previous title did not adequately do so. Regarding this issue, we have modified our title to "New Internationalization Paths of Chinese Brands: A Configurational Study" (lines 1-2) based on our key contributions of (1) synthesizing brand samples across multiple industries and discovering two new paths different from prior research, and (2) examining the complex causality of brand internationalization using fsQCA which revealed interactions among drivers and uncovered market-oriented and culture-oriented patterns previously undiscovered.

2. Response to comment: ("Please reorganize the manuscript at the journal request. Please change the reference format.")

Response: Thank you very much for your meticulous and professional advice. We carefully read the Submission Guidelines of PLoS One to ensure that all parts of the manuscript and each type of content meet the requirements. At the same time, we also noticed that the journal requires the submission of manuscripts to use the ICMJE style of references, so our reference format follows the requirements and adopts the ICMJE style (lines 752-905).

The relevant modifications are in the section References of the manuscript.

3. Response to comment: ("The language of this manuscript is very bad and needs help from native speakers.")

Response: We also strongly agree that the language of the manuscript should be smooth to increase its readability and facilitate readers to correctly understand the content of the manuscript. Considering the limitations of our own English proficiency, we also recognize that we need the help of native English speakers. Therefore, we carefully screened academic article polishing institutions and chose American Journal Experts (AJE), a polishing brand under Springer Nature, to help us improve the language of our manuscript. We hope that the language of the polished manuscript can meet the requirements of the journal. The proof of AJE's completion of polishing is as follows. Due to the order number may disclose personal information, to meet the blind review requirements of the journal, we have mosaic processed the relevant information of the polishing proof.

4. Response to comment: ("Abstracts should include the purpose and findings of the study.")

Response: Thank you very much to the reviewer for providing such detailed and accurate suggestions in Abstract. Our Abstract is indeed not well organized, and we strongly agree that the Abstract section needs to include the purpose and findings of the study so that readers can understand the main research content and findings of the article through the Abstract. Therefore, we have rewritten the abstract according to the purpose and findings of the study. The first half of the Abstract outputs our purpose (lines 8-15) based on the research gap, and the second half of the Abstract summarizes the findings of the study (lines 15-23).

The relevant modifications are in Abstract of the manuscript.

5. Response to comment: ("LINES28-34 This a very vague statement. These sentences do not provide any information on how the concept could be conceptualized?")

Response: Thank you very much for pointing out that our presentation in this section needs further improvement. Lines 28-34 of the original manuscript are a statement about the research gap, which does not provide any information on how the concept could be conceptualized. So we have rewritten the introduction section (lines 25-58) and proposed our research gap based on the gap between the practical and theoretical background (lines 40-46). 

The relevant modifications are in section 1. Introduction of the manuscript.

6. Response to comment: ("LINES69-87. This section should explain the study's context and research objective. Furthermore, the research gap needs to be narrowed after analyzing the previous studies. The research method is not adequately explained in the first section.")

Response: Thank you very much to the reviewers for their detailed suggestions on the Introduction, which have been very helpful for us to improve our manuscript. We strongly agree with the reviewer's suggestion that we should explain the background, research subjects and research methods, and summarize our research gaps based on existing research in this section. We have rewritten the Introduction section of the manuscript according to your suggestion.

("LINES69-87. This section should explain the study's context and research objective. Furthermore, the research gap needs to be narrowed after analyzing the previous studies "): We have rewritten the Introduction section and introduced the study's context based on the practical background of Chinese brand internationalization (lines 26-40) and the theoretical background of Chinese brand internationalization (lines 40-46). Based on the gap between the practical and theoretical background, we have identified our research gap (lines 42-46). In order to better identify our research gap, we have also supplemented the Literature Review (lines 59-158). In addition, we also introduced our research subjects, research issues, and research innovation (lines 47-58).

("The research method is not adequately explained in the first section."): In the original manuscript, we did not provide a very detailed introduction to our research method, which is unfriendly to some readers who are not familiar with this new method. We are also very grateful to the reviewer for carefully pointing out this issue. Therefore, in section 3. Research and Data Methodology, we add subsection 3.2. The Research Method (lines 247-283), which provides a detailed introduction to fsQCA, hoping that the new content can make up for the shortcomings mentioned by the reviewer in our research method introduction.

The relevant modifications are in section 1. Introduction, section 2. Literature Review and subsection 3.2. Research Method of the manuscript.

7. Response to comment: ("Introduction, what authors wanted to convey. Here author must build research gap following the previous studies. The manuscript does not answer the following concerns: Why is it timeliness to explore such a study? What makes this study different from the previously published studies? Are there any similarly findings in line with the previously published studies? Are the findings different from prior academic studies that were conducted elsewhere, if any? For example, Transformation Drive Internationalization, what it requires, what are the new technologies, some recent issue highlights the importance. See the following: New Energy-Driven Construction Industry: Digital Green Innovation Investment Project Selection of Photovoltaic Building Materials Enterprises Using an Integrated Fuzzy Decision Approach. Systems 2023, 11, 11. https://doi.org/10.3390/systems11010011")

Response: Thank you very much to the reviewer for pointing out the issues with our structure in the introduction section, and kindly providing us with corresponding benchmark articles for our reference and learning. In this article, we attach great importance to the impact of digital technology on enterprise investment projects. We are deeply inspired and plan to consider the impact of digital technology on brand internationalization in future research. We cited this literature (lines 746-749) in Directions for Future Research, so that readers can pay attention to the impact of digital technology. In addition, we have made the following modifications in response to the suggestions made by the reviewers in the introduction section.

("Introduction, what authors wanted to convey. Here author must build research gap following the previous studies. "): We have rewritten the Introduction section and identified our research gap (lines 42-46) based on the gap between the practical background of Chinese brand internationalization (lines 26-40) and the theoretical background of Chinese brand internationalization (lines 40-46). In order to better identify our research gap, we have also supplemented the Literature Review (lines 59-158), hoping that the added content can better improve the shortcomings of our original manuscript in proposing research gap.

("The manuscript does not answer the following concerns: Why is it timeliness to explore such a study? What makes this study different from the previously published studies? Are there any similarly findings in line with the previously published studies? Are the findings different from prior academic studies that were conducted elsewhere, if any? For example, Transformation Drive Internationalization, what it requires, what are the new technologies, some recent issue highlights the importance. See the following: New Energy-Driven Construction Industry: Digital Green Innovation Investment Project Selection of Photovoltaic Building Materials Enterprises Using an Integrated Fuzzy Decision Approach. Systems 2023, 11, 11. https://doi.org/10.3390/systems11010011 "): Thank you very much to the reviewer for carefully listing the questions that should be answered in the Introduction section. We have also reorganized our Introduction based on the proposed structure. We answered the question "Why is it timeline to explore such a study?" by examining the practical background of Chinese brand internationalization and the theoretical background of research related to Chinese brand internationalization (lines 26-40). We stated the difference between the practical and theoretical background (lines 40-46) to answer the question: "What makes this study different from the previously published studies?" and "Are there any similarly found in line with the previously published studies?”. In addition, we added the innovative aspects of this study in the Introduction section (lines 50-58), to answer the question "Are the findings different from prior academic studies that were conducted elsewhere, if any?”.

The relevant modifications are in section 1. Introduction and section 2. Literature Review of the manuscript.

8. Response to comment: ("Methodology: Model. I suggest authors here build your main heading on Research and data methodology. Clearly explain the model building process, and what previous studies have used similar models (model testing approach).")

Response: Thank you very much for the constructive suggestions provided by the reviewer on the Research Model, which helped us improve the structure of the article and make it more fluent and readable. Based on the above suggestions, we have carefully made the following modifications.

("Methodology: Model. I suggest authors here build your main heading on Research and data methodology."): We have reorganized the title of this section and highly endorse the reviewer's suggestion to establish a first-level title: 3. Research and Data Methodology, and established the following secondary headings: 3.1. Research Model, 3.2. Research Method, 3.3. Research Data (lines 159-336).

("Clearly explain the model building process, and what previous studies have used similar models (model testing approach). "): In the Research Framework section, we propose the research framework of this article based on the definition of brand internationalization, resource-based view, and institutional theory (lines 162-199). At the same time, based on the boundary spanning theory and referring to relevant research models, we proposed the research model of this article (lines 200-222). 

The relevant modifications are in section 3. Research and Data Methodology of the manuscript.

9. Response to comment: ("There is no flow in the text. It partly depends on the lack of proofreading but also on the fact that many statements and claims are made without being followed up by a clear and logical discussion. It is especially problematic in the Introduction that brings up several findings from different areas without linking them together.")

Response: Thanks to the reviewers for pointing out this problem. We are deeply aware of the issue with the original manuscript. The Introduction of the previous manuscript mixed the content of introduction and literature review, resulting in a relatively rough statement in this section, which is not conducive to readers' better reading of our manuscript. So we have rewritten the Introduction section and added a Literature Review section. In the Introduction section, we have refined it based on practical background, theoretical background, research gap, research content, and research contributions (lines 25-58). In the Literature Review section (lines 59-158), we improved Table 2 (Table 1 of the original manuscript), analyzed the research problems, research subjects, research methods and theoretical support (lines 70-71), and identified our research gap (lines 128-158). We hope the above modifications can address the shortcomings of the original manuscript in language organization and theoretical summary.

The relevant modifications are in section 1. Introduction and section 2. Literature Review of the manuscript.

10. Response to comment: ("Please make sure your conclusions' section underscores the scientific value-added of your paper, and/or the applicability of your findings/results. Highlight the novelty of your study. In addition to summarizing the actions taken and results, please strengthen the explanation of their significance. It is recommended to use quantitative reasoning comparing with appropriate benchmarks, especially those stemming from previous work. See the following: An adoption-implementation framework of digital green knowledge to improve the performance of digital green innovation practices for industry 5.0, https://doi.org/10.1016/j.jclepro.2022.132608.")

Response: Thank you very much to the reviewer for pointing out the issues in the Conclusion section of the manuscript, and providing thoughtful articles for reference. By reading the recommended articles, we have deepened our understanding of the advantages and limitations of fsQCA. We cited this article in section 8. Limitations and Directions for Future Research (lines 738-740), so that readers can continue to understand fsQCA. In addition, the Conclusions section of the original manuscript is quite cumbersome and lacks prominent focus. Therefore, we have rewritten the Conclusions section of the original manuscript using the Discussion-Conclusion-Implications structure proposed by the reviewer. Based on the suggestions provided by the reviewer, we have made the following modifications.

("Please make sure your conclusions' section underscores the scientific value-added of your paper, and/or the applicability of your findings/results. Highlight the novelty of your study. "): We have reorganized the Conclusions section of the manuscript into three parts: (1) Brand Internationalization Modes (lines 629-636), (2) Brand Internationalization Paths (lines 637-660), and (3) Brand Internationalization Capabilities (lines 661-668). Each part presents a comparison of previous research, especially in the key research conclusion of this article. (2) Brand Internationalization Paths section, We have divided into two parts: (i) Differences from Previous Studies (lines 643-655) and (ii) Similarities to Previous Studies (lines 656-660), highlighting the novelty of your study.

(“In addition to summarizing the actions taken and results, please strengthen the explanation of their significance.”): We strongly agree with the reviewer's suggestion to strengthen the explanation of their significance, which will benefit different readers to read the contribution of our article. Therefore, we have added section 7. Implications, and subdivided into two parts: 7.1. Managerial Implications (lines 670-692) and 7.2. Theoretical Implications (lines 693-729), explaining the significance of the article's conclusions from different perspectives.

(“It is recommended to use quantitative reasoning comparing with appropriate benchmarks, especially those stemming from previous work.”): We also strongly agree that comparing with previous studies is beneficial for extracting the innovative aspects of our research. Therefore, we have reorganized the Conclusions section of the manuscript and presented a comparison with previous studies of each point, highlighting the novelty of your study. Additionally, we added subsection 5.1. Theoretical Analysis (lines 403-538) in section 5. Discussions to enhance the novelty of our conclusions by comparing them with previous theories and studies.

The relevant modifications are in section 5. Discussion, section 6. Conclusions and section 7. Implications of the manuscript.

11. Response to comment: ("More importantly, the choice of the variables should be explained in light of the theory and the prior literature on the topic. The arguments are simply relationships and causes very close to the replication of many studies dealing with the same thing. ")

Response: Thank you very much for the reviewer's suggestion that the selection of our variables should be rigorously discussed based on existing literature. This is a constructive suggestion that we strongly agree with. We have rewritten our subsection 3.1.Research Model based on existing theories and research. In 3.1.1. Research Framework (lines 162-199), we propose the research framework based on the definition of brand internationalization, resource-based view, and institutional theory. In 3.1.2. Brand Empowerment Mechanism (lines 200-222), we used boundary spanning theory and combined it with previous research to demonstrate the variables selection, and proposed our research model. In addition, we also summarized the Brand Empowerment Mechanism (lines 223-246) based on brand empowerment theory, boundary spanning theory, and social network theory.

The relevant modifications are in subsection 3.1. Research Model of the manuscript.

12. Response to comment: ("The authors should emphasize the important role of digital technology in future research. See the following: Enhancing Digital Innovation for the Sustainable Transformation of Manufacturing Industry: A Pressure-State-Response System Framework to Perceptions of Digital Green Innovation and Its Performance for Green and Intelligent Manufacturing. https://doi.org/10.3390/systems10030072 ")

Response: Thank you very much to the reviewers for pointing out future research directions on digital technology and providing us with excellent articles that can be referenced. We also strongly agree with this and have plans to consider the impact of digital technology on brand internationalization in future research. We emphasize the importance of considering digital technology in subsequent research in section 8 The Limitations and Directions for Future Research (lines 746-749). In addition, we note that this literature focuses on data empowerment, and the relevant content helps to support the proposed definition of brand empowerment in our article. Therefore, we have cited this article in the definition of brand empowerment section (lines 187-191), so that readers interested in digital technology can have a deeper understanding of empowerment theory.

The relevant modifications are in section 8. Limitations and Directions for Future Research of the manuscript.

13. Response to comment: ("Please consider this structure for manuscript final part: Discussion – Conclusion - Managerial Implication - Practical/Social Implications. Discussion needs to be a coherent and cohesive set of arguments that take us beyond this study in particular, and help us see the relevance of what authors have proposed. Authors should create an independent “Discussion” section. Author need to contextualize the findings in the literature, and need to be explicit about the added value of your study towards that literature. Also other studies should be cited to increase the theoretical background of each of the method used. ")

Response: Thanks to the reviewers for pointing out this problem.

("Please consider this structure for manuscript final part: Discussion-Conclusion -Managerial Implication-Practical/Social Implications."): Following the structure suggested by the reviewer can effectively improve the smoothness of our manuscript's structure, so we reorganized the structure of the manuscript to: 5. Discussion, 6. Conclusion, 7. Implications. We subdivided 7. Implications into 7.1. Managerial Implications and 7.2. Theoretical Implications.

("Discussion needs to be a coherent and cohesive set of arguments that take us beyond this study in particular, and help us see the relevance of what authors have proposed. Authors should create an independent “Discussion” section. Author need to contextualize the findings in the literature, and need to be explicit about the added value of your study towards that literature."): We have added section 5. Discussion, and divided it into subsections: 5.1. Theoretical Analysis (lines 403-538) and 5.2. Case Analysis (lines 539-621). In Theoretical Analysis, we further analyzed the results and extracted paths based on existing theories and research, in order to explain the added value of our study. In addition, we also analyzed the cases covered by each path in Case Analysis, providing practical support for our research results. Comprehensive analysis of two subsections to take us beyond this study in detail.

("Also other studies should be cited to increase the theoretical background of each of the method used."): In the original manuscript, we did not provide a very detailed introduction to our research method, which is unfriendly to some readers who are not familiar with this new method. We are also very grateful to the reviewer for carefully pointing out this issue. Therefore, in section 3. Research and Data Methodology, we added subsection 3.2. Research Method (lines 247-283) to provide a detailed introduction to the theoretical background of fsQCA, hoping that the new content can make up for the shortcomings mentioned by the reviewer in our research method introduction.

The relevant modifications are in subsection 3.2. Research Method, section 5. Discussion, section 6. Conclusion and section 7. Implications of the manuscript.

14. Response to comment: ("As any emprical study that use different approaches I would like to ask to introduce in the Conclusion section at least a paragraph containing the study limitations.")

Response: Thanks to the reviewers for pointing out this problem. We have added section 8. Limitations and Directions for Future Research (lines 730-749), and added the limitations of our research methods (lines 738-740) in Limitations. The exploration of the limitations of research methods is inspired by the following articles: “Yin S, Yu YY. An Adoption-Implementation Framework of Digital Green Knowledge to Improve the Performance of Digital Green Innovation Practices for Industry 5.0. Journal of Cleaner Production 2022;363”, which deepens our understanding of this method. We have cited this article so that readers can continue to understand fsQCA.

The relevant modifications are in section 8. Limitations and Directions for Future Research of the manuscript.

We have tried our best to improve the manuscript and made some changes. The revised portions are highlighted in grey in the paper.

We appreciate the Editors' warm work and the Reviewers' valuable comments and hope that the correction will meet with approval.

Once again, thank you very much for your comments and suggestions.

---

## [Decision Letter · Decision Letter 2]

18 Sep 2023

New Internationalization Paths of Chinese Brands: A Configurational Study

PONE-D-22-33203R2

Dear Dr. Liao,

We’re pleased to inform you that your manuscript has been judged scientifically suitable for publication and will be formally accepted for publication once it meets all outstanding technical requirements.

Kind regards,

J E. Trinidad Segovia

Section Editor

PLOS ONE

Additional Editor Comments (optional):

Reviewers' comments:

Reviewer's Responses to Questions

**Comments to the Author**

1. If the authors have adequately addressed your comments raised in a previous round of review and you feel that this manuscript is now acceptable for publication, you may indicate that here to bypass the “Comments to the Author” section, enter your conflict of interest statement in the “Confidential to Editor” section, and submit your "Accept" recommendation.

Reviewer #3: (No Response)

2. Is the manuscript technically sound, and do the data support the conclusions?

Reviewer #3: (No Response)

3. Has the statistical analysis been performed appropriately and rigorously? 

Reviewer #3: (No Response)

4. Have the authors made all data underlying the findings in their manuscript fully available?

Reviewer #3: (No Response)

5. Is the manuscript presented in an intelligible fashion and written in standard English?

Reviewer #3: (No Response)

6. Review Comments to the Author

Reviewer #3: I am satisfied with the revisions carried out based on earlier feedback. The paper is in need of a final language check, preferably by an experienced or professional proofreader, to improve the clarity of expression and impact of your ideas. Once this is resolved, your paper will be ready for acceptance.

7. PLOS authors have the option to publish the peer review history of their article (what does this mean?). If published, this will include your full peer review and any attached files.

Reviewer #3: No

---

## [Editor Report · Acceptance letter]

22 Sep 2023

PONE-D-22-33203R2 

New Internationalization Paths of Chinese Brands: A Configurational Study 

Dear Dr. Liao:

I'm pleased to inform you that your manuscript has been deemed suitable for publication in PLOS ONE. Congratulations! Your manuscript is now with our production department. 

Kind regards, 

on behalf of

Dr. J E. Trinidad Segovia 

Section Editor

PLOS ONE